# Emergent Analogical Reasoning in Transformers

**Gouki Minegishi** [1]  **Jingyuan Feng** [1]  **Hiroki Furuta** [2] [*]  **Takeshi Kojima** [1]  **Yusuke Iwasawa** [1]  **Yutaka Matsuo** [1]

## Abstract

Analogy is a central faculty of human intelligence, enabling abstract patterns discovered in one domain to be applied to another. Despite its central role in cognition, the mechanisms by which Transformers acquire and implement analogical reasoning remain poorly understood. In this work, inspired by the notion of functors in category theory, we formalize analogical reasoning as the inference of correspondences between entities across categories. Based on this formulation, we introduce synthetic tasks that evaluate the emergence of analogical reasoning under controlled settings. We find that the emergence of analogical reasoning is highly sensitive to data characteristics, optimization choices, and model scale. Through mechanistic analysis, we show that analogical reasoning in Transformers decomposes into two key components: (1) geometric alignment of relational structure in the embedding space, and (2) the application of a functor within the Transformer. These mechanisms enable models to transfer relational structure from one category to another, realizing analogy. Finally, we quantify these effects and find that the same trends are observed in pretrained LLMs. In doing so, we move analogy from an abstract cognitive notion to a concrete, mechanistically grounded phenomenon in modern neural networks.

## 1. Introduction

Recent years have witnessed remarkable progress in the reasoning capabilities of large language models (LLMs), particularly in constructing chains of intermediate reasoning before the final answer (Wei et al., 2022; Kojima et al., 2022; OpenAI et al., 2024; Google DeepMind, 2025; DeepSeek-AI et al., 2025). These developments have renewed interest in a key question: *how do LLMs realize reasoning?*

Much of recent research on understanding reasoning frames reasoning as *compositional reasoning*, where complex reasoning arises from sequentially composing simpler primitives. For example, given the facts

(i) Alice is Bob's mother $(a \to b)$,

(ii) Bob is Carol's father $(b \to c)$,

LLMs can infer that Alice is Carol's grandmother $(a \to c)$ by composing two known relations (Yang et al., 2018; Mavi et al., 2024). The mechanisms underlying this form of reasoning have been widely studied, including its emergence during training (He et al., 2024), its dependence on data structure (Wang et al., 2024; Schug et al., 2025) and its scaling behavior (Petty et al., 2024; Redhardt et al., 2025).

Beyond compositional reasoning, humans exhibit a qualitatively different form of reasoning, *analogy*. Rather than producing conclusions by chaining local steps, analogy identifies shared relational structure across distinct domains, enabling a form of "leap" (Gentner, 1983; Holyoak & Thagard, 1996; Bartha, 2013). A classic example from cognitive science is the analogy between the solar system and atomic structure (Gentner, 1983), where each domain consists of three entities and their relations:

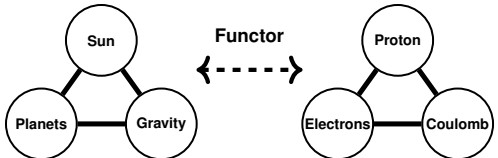

One can infer a correspondence between entities across domains, such as mapping the **Sun** to the **Proton**. This inference does not arise from the entities' intrinsic similarity. Instead, it emerges from the similarity of entities' relational roles within each domain. Thus, analogical reasoning can be viewed as operating on *relations between relations*, rather than on relations among individual entities. In category theory, this can be formalized as a mapping between categories[1], namely a *functor* (Awodey, 2010). This ability is widely regarded as a central faculty of human intelligence, enabling efficient learning from limited experience (Thagard, 1992; Gentner & Hoyos, 2017) and is often viewed as a source of creativity and science discovery (Leatherdale, 1974; Goel, 1997; Gentner et al., 1997).

---

[*]Work done as an advisory role only. [1]The University of Tokyo [2]Google Deep Mind. Correspondence to: Gouki Minegishi <minegishi@weblab.t.u-tokyo.ac.jp>.

*Proceedings of the 43rd International Conference on Machine Learning*, Seoul, South Korea. PMLR 306, 2026. Copyright 2026 by the author(s).

---

[1]Here, we use *category* as a formal abstraction of a *domain*, consisting of entities and their relations.

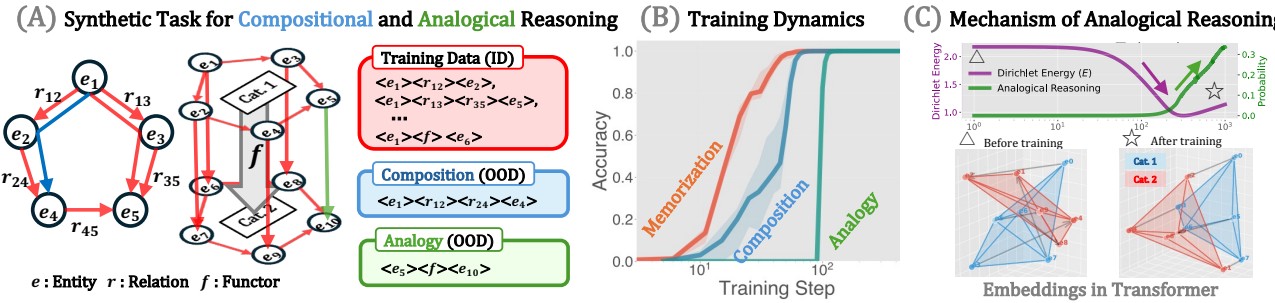

*Figure 1.* **(A) Synthetic task for compositional and analogical reasoning.** Compositional reasoning evaluates whether a model can combine facts observed in-distribution (ID) during training to infer novel combinations (out-of-distribution, OOD). Analogical reasoning assesses whether a mapping $f$ (functor) between distinct categories generalizes. Solving analogical reasoning requires capturing the underlying relational structure of each category from the ID facts. **(B) Training dynamics of Transformer.** When training a Transformer on this task, the model first fits on in-distribution data, then acquires compositional reasoning, and finally succeeds at analogical reasoning. **(C) Mechanism of analogical reasoning.** We analyze internal representations of the Transformer before and after the emergence of analogical reasoning. After acquiring the ability for analogical reasoning, the model develops a well-structured embedding space, which is quantitatively characterized by a decrease in Dirichlet Energy.

Despite its long-standing significance in intelligence, it remains unclear *when* and *how* Transformer-based architectures acquire analogical reasoning. While several works probe analogical performance at the behavioral level (Chen, 2022; Webb et al., 2023; Ye, 2024; Yasunaga et al., 2024; Johnson et al., 2025), we lack a systematic understanding.

In this work, we take a step toward filling this gap. Inspired by the notion of *functor* in category theory, we formalize analogical reasoning as inferring correspondences across categories. Based on this formulation, we design synthetic tasks to evaluate *compositional* and *analogical reasoning* within a unified framework (Figure 1-(A)). Our task is based on atomic facts provided in the in-distribution (ID) training data, where each fact specifies a relation ($r_{s \to t}$) between a pair of entities ($e_s, e_t$). In compositional reasoning, we test whether a model can combine learned atomic facts to infer novel combinations (out-of-distribution, OOD). In analogical reasoning, we consider two categories that share the same relational structure but differ in their entities. The model is required to infer the corresponding entity across categories based on their relational roles. Since evaluation for analogical reasoning is also performed in OOD, the model must capture the underlying relational structure of each category from the ID facts.

Using this synthetic task, we analyze *when* compositional and analogical reasoning emerges during training. We observe a clear three-stage learning dynamics (Figure 1-(B)): models first fit in-distribution facts, then acquire compositional reasoning, and later develop analogical reasoning. We find that, unlike compositional reasoning, the emergence of analogical reasoning is highly sensitive to data characteristics and optimization settings (e.g., weight decay) and does not improve monotonically with model size. This suggests that analogical reasoning relies on qualitatively different mechanisms from compositional reasoning, and that these

mechanisms cannot be explained solely by weight-norm regularization or by increasing model capacity.

Motivated by these findings, we further investigate *how* analogical reasoning is implemented mechanistically in Transformer. We show that analogical reasoning can be decomposed into two components: (1) structural alignment in the embedding space and (2) functor application in Transformer layers. In the synthetic task, analogical reasoning emerges after embeddings of entities across categories become geometrically aligned (Figure 1-(C)), which can be measured by a substantial decrease in Dirichlet Energy during training. This alignment is subsequently exploited by Transformer to transform a source entity ($e_s$) into its analogical counterpart ($e_t$), with the functor ($f$) being applied as a vector addition ($e_t \approx e_s + f$). Furthermore, we probe pretrained LLMs using in-context learning and observe similar signatures. While in the synthetic task, the decrease in Dirichlet Energy occurs along the training-step axis, in LLMs, the same phenomenon unfolds along the layer axis. These results indicate that the analogical reasoning mechanism discovered in the synthetic task is also present in pretrained LLMs.

Unlike recent reasoning approaches that emphasize chaining local steps of thought, analogical reasoning enables conceptual leaps across domains. As such, it offers the basis for a distinct reasoning paradigm beyond sequential composition. We hope that our work provides a foundation for studying analogy in Transformers.

We organize the paper as follows. In **Section 2**, we propose a synthetic task designed to evaluate both compositional and analogical reasoning. In **Section 3**, we present a detailed analysis of training dynamics in Transformers on this task. In **Section 4**, we show the mechanistic implementation of analogical reasoning in Transformers, and in **Section 5**, we further demonstrate that analogous mechanistic signatures are also present in pretrained LLMs.

## 2. Synthetic Task for Analogical Reasoning

We propose a synthetic task to evaluate compositional and analogical reasoning. The task is defined over entities and relations and consists of three types of facts: atomic, compositional, and analogical facts.

### 2.1. Problem Formulation

**Entities and Relations.** Let $\mathcal{E}$ denote a finite set of entities and $\mathcal{R}$ a finite set of relations. We partition the entity set into two disjoint subsets $(\mathcal{E}_1, \mathcal{E}_2)$,[2] which correspond to two categories in Figure 1. On $\mathcal{E}_1$, we construct a directed complete graph whose edges are labeled by relations. Formally, for each ordered pair $(e_i, e_j) \in \mathcal{E}_1 \times \mathcal{E}_1$ with $e_i \neq e_j$, we assign a relation label $r(e_i, e_j) \in \mathcal{R}$, sampled uniformly at random from $\mathcal{R}$, with the constraint that each entity $e_i \in \mathcal{E}_1$ has distinct relation labels on its outgoing edges.[3]

**Atomic facts.** An *atomic fact* represents a single labeled edge in the relational graph on $\mathcal{E}_1$, and is given by the triple

$$(e_s,\ r(e_s, e_t),\ e_t) \in D_{\text{atomic}}$$

We denote by $\mathcal{D}_{\text{atomic}}$ the set of atomic facts. Atomic facts constitute the basic relational knowledge during training.

**Compositional facts.** From atomic facts, we derive *compositional facts* that correspond to two-hop relational compositions. A compositional fact is defined as the quadruple

$$(e_s,\ r(e_s, e_i),\ r(e_i, e_t),\ e_t) \in D_{\text{comp}}$$

which is obtained by composing the following two atomic facts that share the intermediate entity $e_i$: $(e_s, r(e_s, e_i), e_i)$ and $(e_i, r(e_i, e_t), e_t)$. We denote by $\mathcal{D}_{\text{comp}}$ the set of compositional facts.

**Analogical facts.** To formalize analogy across categories, we consider a bijection $\mathcal{F} : \mathcal{E}_1 \to \mathcal{E}_2$, which induces a one-to-one correspondence between entities in the two categories. We transfer the relational structure from $\mathcal{E}_1$ to $\mathcal{E}_2$ by defining $r(\mathcal{F}(e_s), \mathcal{F}(e_t)) = r(e_s, e_t), \forall e_s \neq e_t \in \mathcal{E}_1$. As a result, $\mathcal{E}_1$ and $\mathcal{E}_2$ share a relational structure. From a category-theoretic perspective (Awodey, 2010), this mapping $\mathcal{F}$ can be viewed as a *functor*. An *analogical fact* states this cross-category alignment as the triple

$$(e_s,\ f,\ \mathcal{F}(e_s)) \in D_{\text{analogical}},$$

where $e_s \in \mathcal{E}_1$ and $\mathcal{F}(e_s) \in \mathcal{E}_2$, and $e_s \neq \mathcal{F}(e_s)$ since the two sets are disjoint. We denote by $\mathcal{D}_{\text{analogical}}$ the set of analogical facts. Here, $f$ is treated as a special symbol.

---

[2] $\mathcal{E} = \mathcal{E}_1 \cup \mathcal{E}_2, \mathcal{E}_1 \cap \mathcal{E}_2 = \varnothing, |\mathcal{E}_1| = |\mathcal{E}_2|$

[3] If an entity has multiple outgoing edges with the same relation and is used as the intermediate node in a compositional fact, compositional reasoning becomes impossible.

*Table 1.* Tokenization for each fact.

| Component | Token Representation |
|---|---|
| Atomic fact (3 tokens) | $\langle e_s \rangle \langle r(e_s, e_t) \rangle \langle e_t \rangle$ |
| Compositional fact (4 tokens) | $\langle e_s \rangle \langle r(e_s, e_i) \rangle \langle r(e_i, e_t) \rangle \langle e_t \rangle$ |
| Analogical fact (3 tokens) | $\langle e_s \rangle \langle f \rangle \langle \mathcal{F}(e_s) \rangle$ |

**Compositional and Analogical reasoning.** We now define the two types of generalization evaluated in our task: compositional and analogical reasoning. While both require extrapolation beyond the training data, they rely on qualitatively different capabilities.

**Definition 2.1** (Compositional reasoning). Let $\mathcal{D}_{\text{comp}}^{\text{OOD}}$ be a held-out set of compositional facts such that it contains constituent atomic fact, but the composed quadruple itself does not. A model is said to exhibit *compositional reasoning* if it can correctly predict the final entity in samples from $\mathcal{D}_{\text{comp}}^{\text{OOD}}$, given the preceding entity and two relation tokens.

**Definition 2.2** (Analogical reasoning). Let $\mathcal{D}_{\text{analogical}}^{\text{OOD}}$ be a held-out set of analogical facts. A model is said to exhibit *analogical reasoning* if it can correctly predict the counterpart entity $\mathcal{F}(e)$ in $\mathcal{E}_2$ from the prefix $(e, f)$, despite the fact that the corresponding triple is not observed during training.

Compositional reasoning primarily requires the ability to combine acquired relational knowledge to infer novel outcomes. In contrast, analogical reasoning demands learning the underlying relational structure of each category and leveraging this structure to generalize.

### 2.2. Experiment Setup

**Dataset.** The dataset is characterized by the following controllable configurations: (1) the total number of entities $|\mathcal{E}|$, (2) the number of relations $|\mathcal{R}|$, and (3) the OOD ratio for compositional facts ($|D_{\text{comp}}^{\text{OOD}}|/|D_{\text{comp}}|$) and (4) for analogical facts ($|D_{\text{analogical}}^{\text{OOD}}|/|D_{\text{analogical}}|$). Unless otherwise specified, we use the following default configuration: $|\mathcal{E}| = 20$ entities in total, $|\mathcal{R}| = 10,000$ relations, and an OOD ratio of $0.1$ for both compositional and analogical facts. The vocabulary consists of $|\mathcal{E}|$ entity tokens, $|\mathcal{R}|$ relation tokens, and a single functor token, yielding a total vocabulary size of $|\mathcal{E}| + |\mathcal{R}| + 1$. Concrete examples of each fact type and their tokenized representations are summarized in Table 1.

**Model and Training setup.** Following prior work on synthetic tasks (Chan et al., 2022; Reddy, 2024; Minegishi et al., 2025), we train models using a cross-entropy loss applied only to the final token of each sequence. Our default model is a causal Transformer with a single layer and a single attention head, with a dimension of 128. We use the Adam optimizer (Kingma, 2014) with a learning rate of $10^{-4}$, weight decay set to $0$, and a batch size of $32$. All reported results are averaged over three random seeds. Additional implementation details are provided in Appendix A.

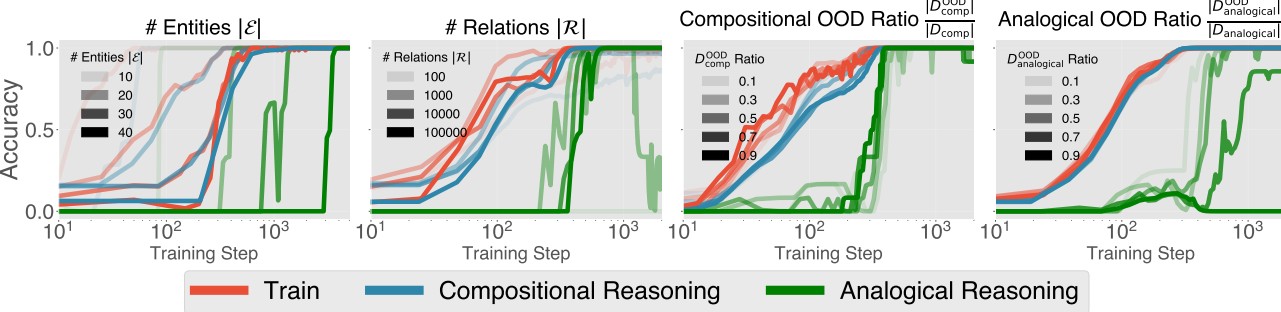

*Figure 2.* Learning dynamics under varying data properties in **training data**, **compositional** and **analogical** reasoning. From left to right, we vary (i) the number of entities ($|\mathcal{E}|$), (ii) the number of relations ($|\mathcal{R}|$), (iii) the compositional OOD ratio, and (iv) the analogical OOD ratio. While training accuracy and compositional generalization improve smoothly across settings, analogical reasoning consistently emerges later and exhibits unique behavior.

## 3. Emergent Analogical Reasoning

We first examine the learning dynamics of a 1-layer Transformer on our synthetic task[4]. As shown in Figure 1-(B) for the case of 10 entities, training exhibits a clear *three-stage progression*. The model initially fits the in-distribution data (**memorization**), then acquires **compositional reasoning**, and later develops **analogical reasoning**. Accordingly, we analyze this emergence from three perspectives: data characteristics (Section 3.1), optimization (Section 3.2), and model scaling (Section 3.3).

### 3.1. Data Characteristics Drive Analogical Reasoning

We first investigate how dataset characteristics affect the emergence of analogical reasoning. Figure 2 summarizes the learning dynamics of training accuracy, compositional reasoning and analogical reasoning as we vary four key factors: the number of entities $|\mathcal{E}|$, the number of relations $|\mathcal{R}|$, the compositional and the analogical OOD ratio.

Across all settings, training accuracy improves smoothly, indicating that the model can reliably memorize in-distribution facts. As the number of entities or relations increases, training converges more slowly, reflecting the increased task complexity. Compositional reasoning closely follows the training accuracy. In more complex settings, the gap between training accuracy and compositional reasoning disappears, as memorization itself becomes increasingly difficult. In contrast, analogical reasoning exhibits different behavior. As the number of entities increases, the time required to acquire analogical reasoning grows substantially relative to compositional reasoning. This suggests that analogical reasoning depends on learning underlying relational structures that become harder to capture as the entity set grows.

The number of relations also plays a critical role in ana-

---

[4]Implementation code is available at `https://github.com/gouki510/Analogy_in_Transformer`.

logical reasoning. When the relation set is too small (e.g., $|\mathcal{R}| = 100$), analogical reasoning fails to emerge. This is consistent with the view that analogical reasoning infers entities based on their relational roles, making relational diversity essential for distinguishing and representing entities. Interestingly, in some settings with a very large number of relations (e.g., $|\mathcal{R}| = 1,000$), analogical reasoning is acquired but later lost, exhibiting *transient behavior*, which has been reported on in-context learning works (Park et al., 2024; Singh et al., 2025). We further analyze this phenomenon in the Appendix C. Additionally, increasing the OOD ratio reduces the amount of informative training signal, making generalization more difficult. Higher compositional OOD ratios delay the emergence of compositional reasoning, consistent with prior findings (He et al., 2024; Redhardt et al., 2025). Analogical reasoning is more sensitive to the analogical OOD ratio: when this ratio is high (e.g., 0.9), analogical generalization fails to emerge, highlighting its intrinsic difficulty. More broadly, this data-dependence is in line with prior evidence that distributional properties of the training data can drive (or suppress) the emergence of complex behaviors in Transformers, such as emergent in-context learning (Chan et al., 2022). Notably, the compositional and analogical facts do not interfere with each other: the compositional OOD ratio has little effect on analogical reasoning, and vice versa. This decoupling highlights that analogical reasoning constitutes a qualitatively distinct form of compositional reasoning. We further examine the effect of graph sparsity in Appendix D.

### 3.2. Role of Optimization in Analogical Reasoning

We find that optimization choices (weight decay, batch size, and learning rate) also play a critical role in the acquisition of analogical reasoning. The results are summarized in Figure 3. We first examine the effect of weight decay, which is commonly understood as a mechanism for suppressing memorization. As the weight decay coefficient increases

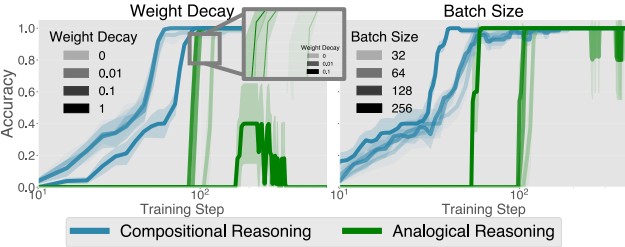

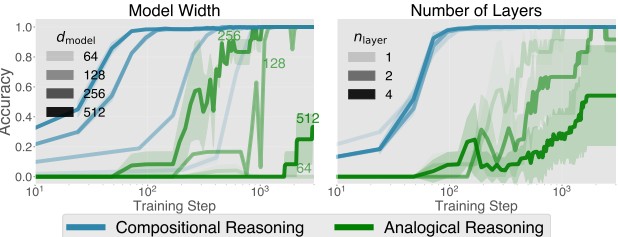

*Figure 3.* Effect of optimization hyperparameters in compositional and analogical reasoning. **Left:** varying weight decay. Moderate weight decay (e.g., 0.01 and 0.1) accelerates the analogical reasoning, whereas excessively strong weight decay (1.0) prevents it, despite compositional reasoning remaining successful. **Right:** varying batch size. Larger batch sizes generally lead to faster acquisition of analogical reasoning.

*Figure 4.* Effect of model scaling on compositional and analogical reasoning. **Left:** accuracy curves for different model widths ($d_{\text{model}}$). **Right:** accuracy curves for different numbers of layers ($n_{\text{layer}}$). Compositional reasoning improves consistently with increasing model size, whereas analogical reasoning shows non-monotonic and unstable scaling behavior.

from 0 to 0.01 and 0.1, analogical reasoning emerges earlier during training. However, when weight decay is set too large (e.g., 1), analogical reasoning fails to be learned. In contrast, compositional reasoning remains robust even under strong weight decay. The role of weight decay in improving generalization has been extensively studied in the context of grokking (Power et al., 2022; Liu et al., 2022; 2023). Prior work has argued that strong norm-based regularization, such as weight decay, shrinks model weights and guides optimization toward more generalizable solutions in the loss landscape. However, our results suggest that the acquisition of analogical reasoning cannot be explained solely by such weight-norm effects, and may require more structured internal representations than those induced by simple norm shrinkage. We also observe that increasing the batch size generally accelerates learning, consistent with standard optimization intuition. Results for learning rate sweeps are provided in Appendix E.

### 3.3. Scaling Behavior of Analogical Reasoning

We investigate how model scaling affects compositional and analogical reasoning by sweeping both model width ($d_{\text{model}}$) and the number of layers ($n_{\text{layer}}$), as shown in Figure 4. Across all settings, compositional reasoning consistently improves with increasing model size. Wider models achieve higher accuracy earlier in training. This scaling behavior aligns with prior findings (Redhardt et al., 2025) that compositional generalization benefits from model scaling.

In contrast, analogical reasoning exhibits different characteristics in scaling. Increasing model size does not monotonically improve performance, and in some cases even degrades it. For example, models with $d_{\text{model}} = 64$ almost never succeed at analogical reasoning. Moderately sized models ($d_{\text{model}} = 128$ and 256) are more likely to acquire analogical reasoning, whereas further increasing the width to 512 makes analogical reasoning more difficult to learn. For depth scaling, we observe that deeper models can under-

perform under our default (fixed) optimization settings. We caution that this behavior can reflect an optimization artifact rather than an architectural limitation: a simple learning-rate sweep recovers strong performance for deeper models (Appendix B, Figure 11). More broadly, this highlights that analogical reasoning is substantially more sensitive to optimization and hyperparameter choices than compositional reasoning. In the next section, we analyze the internal mechanisms underlying its emergence during training.

## 4. The Mechanism of Analogy in Transformer

We next examine how Transformer models implement analogical reasoning mechanistically. We consider analogical mappings of the form

$$e_t = \mathcal{F}(e_s),$$

with $e_s \in \mathcal{E}_1$ and $e_t \in \mathcal{E}_2$. Our analysis decomposes the realization of analogy into two components, illustrated in Figure 5.

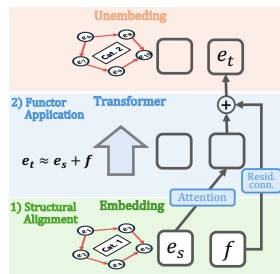

*Figure 5.* Analogical reasoning decomposes into (1) Structural alignment in the embedding, (2) Functor application in Transformer.

**(1) Structural alignment in the embedding space.** The relational structure of each category is captured in the embedding space. **(2) Functor Application.** Attention mechanisms enable the functor token $f$ to retrieve information about the source entity $e_s$. Specifically, $f$ attends to $e_s$ and writes information about $e_s$ into the representation at the position of $f$. Residual connections integrate the retrieved information with the representation of $f$. When the embedding space is well structured as in (1), this integration realizes a vector arithmetic of the form,

$$e_t \approx e_s + f,$$

allowing the model to predict the correct target entity. In the following subsections, we quantify structural alignment

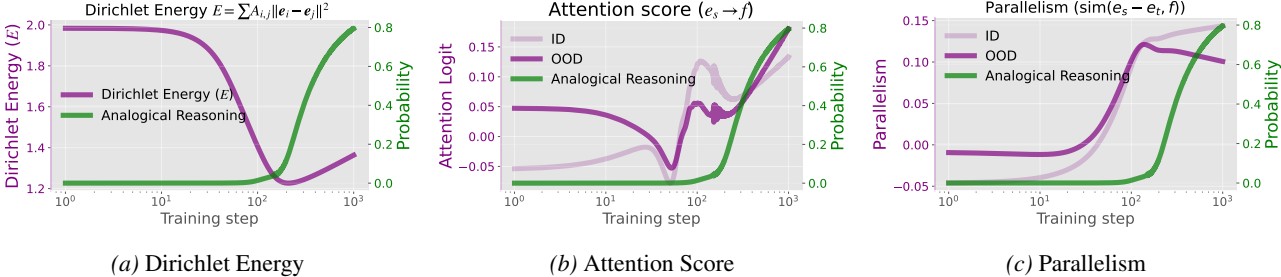

*(a)* Dirichlet Energy        *(b)* Attention Score        *(c)* Parallelism

*Figure 6.* Mechanistic signals underlying the emergence of analogical reasoning, where is measured by the model's probability of the correct target entity ($e_t$). **(a) Dirichlet energy** decreases during training, indicating increasing structural alignment in the embedding space. **(b) Attention Score** from the functor token $f$ to the source entity $e_s$ increase as analogical reasoning emerges, reflecting attention-based information retrieval. **(c) Parallelism**, defined by the similarity between ($e_t - e_s$) and $f$, increases concurrently, indicating that the model realizes analogical reasoning by adding the functor representation $f$ to the source entity embedding $e_s$ via a residual connection.

using Dirichlet Energy (Section 4.1), analyze the implementation of functor application in Transformer (Section 4.2).

## 4.1. Structural Alignment in Embedding Layer

We begin by analyzing embedding-level structural alignment, corresponding to component (1) in Figure 5. As a first step, we visualize entity embeddings[5] before and after the model acquires analogical reasoning. Figure 7 shows PCA projections of entity embeddings from category $\mathcal{E}_1$ (blue) and $\mathcal{E}_2$ (red), with black arrows indicating functor across categories. Before training, embeddings from the two categories are not structurally aligned. After the analogical reasoning, the two categories exhibit clear geometric alignment. We further visualize the training dynamics of entity embeddings using PCA in Appendix F. To quantify this observation, we measure the *Dirichlet Energy* of entity embeddings with respect to a graph defined by our task structure. Let $\boldsymbol{A} \in \mathbb{R}^{|\mathcal{E}| \times |\mathcal{E}|}$ denote the adjacency matrix, and let $\boldsymbol{h}_{e_i} \in \mathbb{R}^d$ denote the embedding of $e_i$. Since our focus is on analogical reasoning, we construct $\boldsymbol{A}$ such that $\boldsymbol{A}_{ij} = 1$ if entities $i$ and $j$ are related via the functor mapping, and $\boldsymbol{A}_{ij} = 0$ otherwise. Following prior work (Park et al., 2025), the Dirichlet Energy is defined as,

$$E(\mathcal{E}) = \sum_{e_i, e_j \in \mathcal{E}} \boldsymbol{A}_{ij} \|\boldsymbol{h}_{e_i} - \boldsymbol{h}_{e_j}\|^2. \tag{1}$$

We provide a detailed derivation of the multi-dimensional formulation in Appendix G. Lower Dirichlet Energy indicates that relationally connected entities are embedded closer together, reflecting increased structural organization in the embedding space. Because Dirichlet Energy depends on representation distances, it can also be influenced by the growth of embedding norms during training; see Appendix S. Figure 6-(a) shows the Dirichlet Energy and the model's probability of predicting the correct target entity

---

[5]Concretely, the entity embedding is the vector corresponding to entity in the embedding matrix.

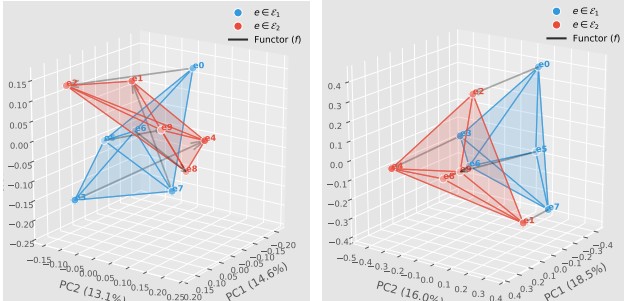

*(a)* Before analogical reasoning    *(b)* After analogical reasoning

*Figure 7.* PCA Visualization of entity embeddings before (0 step) and after ($10^3$ step) the acquisition of analogical reasoning. Entity embeddings from category $\mathcal{E}_1$ (blue) and $\mathcal{E}_2$ (red) are shown, with arrows indicating the functor. **(a) Before**, embeddings from the two categories are not structurally aligned. **(b) After**, the two categories exhibit clear geometric alignment. Results across training epochs are provided in Appendix F.

during training. Analogical performance improves after the Dirichlet Energy has substantially decreased, suggesting that embedding-level structural alignment precedes the emergence of analogical reasoning. The effect of model scaling is discussed in Appendix H.

## 4.2. Functor Application in Transformer

Given the emergence of a structured embedding space (Figure 7), how does the model perform analogical reasoning to infer the corresponding entity? As we show below, this is achieved by *applying the functor as a vector addition* within Transformer layers.

Concretely, the input is ($e_s$, $f$) with $e_s \in \mathcal{E}_1$, and the target is $e_t \in \mathcal{E}_2$. To realize the mapping ($e_s \rightarrow e_t$), two mechanisms are required, corresponding to components (2) in Figure 5. First, through attention, the functor token $f$ retrieves information about the source entity $e_s$ and incorporates it into its own representation. Second, via residual connections, the representation of $f$ is additively integrated with

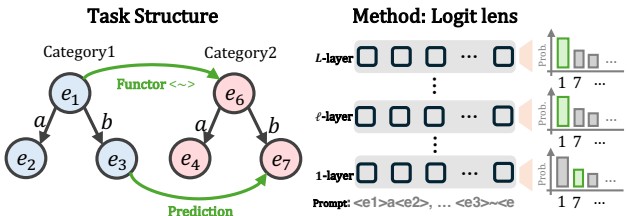

*Figure 8.* **Left:** Task structure for analogical reasoning in LLM experiments. Two categories share the same relational structure defined by relations $a$ and $b$. **Right:** Method overview. We apply the logit lens at each Transformer layer to track how the probability of the correct target entity evolves across layers.

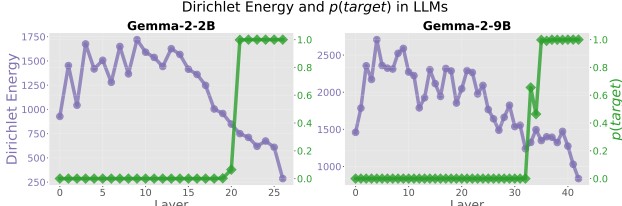

*Figure 9.* Layer-wise evolution of Dirichlet Energy and target probability in LLMs. Dirichlet Energy (purple) and the logit-lens probability of the target entity (green) are shown across layers for GEMMA2-2B (left) and GEMMA2-9B (right). In both models, the Dirichlet energy decreases in the later layers, coinciding with a rapid increase in the probability of the correct target.

that of $e_s$, resulting in a representation that approximates the target entity. Together, these operations implement analogical reasoning as a simple vector addition, $e_t \approx e_s + f$. The first mechanism can be verified by examining attention scores from $f$ to $e_s$. Specifically, we measure the attention weight from the source entity token to the functor token,

$$\text{Attn}(e_s \to f), \qquad (2)$$

where the value is taken from the attention map. As shown in Figure 6-(b), this attention score (purple) tends to increase *prior to* observable improvements in analogical reasoning performance. This is consistent with the view that internal circuit formation or mechanistic progress signals can precede changes in model behavior, as observed in prior work on grokking progress measures (Nanda et al., 2023). This trend suggests that the model increasingly transfers source-entity information to the functor before reliably producing correct analogical outputs. The second mechanism is captured by a measure of geometric parallelism, defined as the cosine similarity,

$$\cos\bigl(\boldsymbol{h}'_t - \boldsymbol{h}_s, \ \boldsymbol{h}_f\bigr), \qquad (3)$$

where $\boldsymbol{h}_s$ denotes the embedding of the source entity, $\boldsymbol{h}_f$ denotes the embedding of the functor token, and $\boldsymbol{h}'_t$ denotes the unembedding of the target entity[6]. This measure evaluates whether the functor representation corresponds to the displacement from the source entity $e_s$ to the target entity $e_t$ in representation space. As shown in Figure 6-(c), this parallelism measure (purple) increases concurrently with analogical reasoning performance. This trend holds for both in-distribution and out-of-distribution settings.

As illustrated in Figure 5, these results demonstrate that when performing analogical reasoning, the model acquires structured representations within each category in the embedding space, and then applies the functor within Transformers to map the source entity to the correct target.

---

[6]Concretely, $\boldsymbol{h}'_t$ corresponds to the vector in the unembedding matrix associated with entity $t$. We use the unembedding representation because the model computes output probabilities over target entities via the final unembedding matrix.

## 5. The Mechanism of Analogy in LLMs

A key question for interpretability (Bereska & Gavves, 2024; Sharkey et al., 2025; Zhang et al., 2026) is whether mechanisms discovered in toy settings align with the behavior of real LLMs. We therefore investigate whether the mechanism of analogical reasoning identified in our toy settings also emerges in real LLMs.

**Method.** We conduct experiments using GEMMA2-2B/9B (Gemma Team, 2024). We probe analogical reasoning in LLMs through *in-context learning*. An overview of our method is shown in Figure 8. We provide the model with the following prompt, where `<e>` denotes entity tokens, `a` and `b` denote relation tokens, and `˜` denotes a functor:

```
<e1>a<e2>, <e1>b<e3>.   <e6>a<e4>,
<e6>b<e7>.   <e1>˜<e6>,  <e3>˜<e
```

where the correct answer is `7`. This prompt closely mirrors the synthetic task (Figure 1): two categories share the same relational structure defined by relations $a$ and $b$, and the model is required to infer the corresponding entity across categories. We intentionally avoid using tokens such as $\langle f \rangle$ and $\langle r_{12} \rangle$, which are the same representations as the synthetic task (Table 1). Because LLMs are pretrained, surface forms such as the string "e1" in entity tokens or relation string "r12" may already carry unintended priors. Similarly, a naive categorical construction (e.g., using entities set $\{e_1, e_2, e_3\}$ for Category 1 and $\{e_4, e_5, e_6\}$ for Category 2) could implicitly encode simple arithmetic patterns such as a "+3" offset as a functor. To avoid these artifacts, we design the prompt so that analogical reasoning is evaluated purely through in-context learning, rather than through pretrained semantic or numerical biases. Results for alternative prompt designs are reported in Appendix I. Because in-context learning unfolds across layers, we apply the logit lens (nostalgebraist, 2020) at the last token in every layer to track how strongly the model predicts the target `7` (Figure 8, right). Since logit lens can underestimate information present in intermediate representations, we additionally compare it with

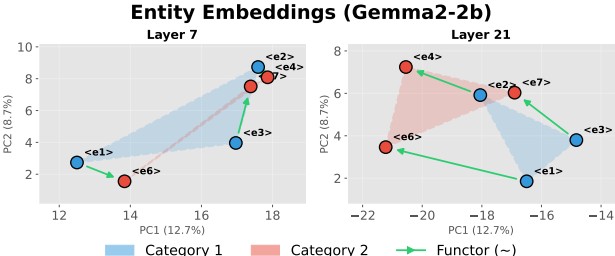

*Figure 10.* PCA projections of entity hidden states at layer 7 (left) and layer 21 (right) in GEMMA2-2B. At layer 7, before the decrease in Dirichlet Energy, entity representations across categories are weakly aligned. By layer 21, after the energy drop, entity representations become geometrically aligned.

a trained linear probe in Appendix R. In addition, following Section 4.1, we analyze the structural organization of hidden states using Dirichlet Energy computed over an adjacency matrix $\boldsymbol{A}$ that connects entities related by the functor: $E(\boldsymbol{H}^\ell) = \sum_{i,j} \boldsymbol{A}_{ij} \|\boldsymbol{h}_{e_i}^\ell - \boldsymbol{h}_{e_j}^\ell\|^2$. Here, $\boldsymbol{H}^\ell \in \mathbb{R}^{T \times d}$ denotes the hidden states at layer $\ell$, taken as the output of the Transformer block at that layer, where $T$ is the number of tokens and $d$ is the hidden dimension, and $\boldsymbol{h}_{e_i}^\ell$ corresponds to the hidden state of the $\langle e_i \rangle$ token. For GEMMA2 models, entity markers such as $\langle e_i \rangle$ are tokenized into multiple sub-tokens (e.g., $<, \ominus, i, >$); we average their hidden states when computing energies.

**Results.** Figure 9 shows the layer-wise Dirichlet Energy (purple) alongside the target's probability by the logit lens (green). For both GEMMA2-2B and GEMMA2-9B, the energy begins to decrease in the later layers, and this decrease is closely accompanied by a corresponding increase in the probability of the correct answer. This indicates that structural alignment between functor-related entities emerges progressively as the model approaches the output layers. Interestingly, while analogical behavior in the toy model appeared along the *training-step axis*, the same phenomenon manifests along the *depth (layer) axis* in LLMs. This suggests that, even without explicit weight updates, LLMs refine their representations across layers, progressively aligning geometric structure toward the output. This behavior is closely related to prior observations (Von Oswald et al., 2023; Deutch et al., 2024) that in-context learning can induce gradient descent-like effects during inference. We further verify that this trend is robust to increasing the number of entities (Appendix J) and is not specific to the GEMMA family, with similar results observed in LLAMA models (Grattafiori et al., 2024) (Appendix K)

Figure 10 visualizes PCA projections of the hidden states of GEMMA2-2B at layer 7 (before the energy decrease) and layer 21 (after the energy decrease). Consistent with our toy experiments, structurally aligned representations become clearly organized only after the energy drops. PCA visualizations for all layers are provided in Appendix L.

## 6. Related Works and Discussion

**Summary.** In this work, we focus on *analogy*, an under-explored aspect of reasoning in LLMs. We introduce a synthetic task designed to analyze analogical reasoning in Section 2. In Section 3, we investigate its training dynamics and their relationship to data characteristics, optimizers, and model scale. We then discover the internal mechanisms in Section 4, and show that closely related mechanisms can also be observed in pretrained LLMs in Section 5. We approach reasoning from a novel perspective based on analogy, while remaining closely connected to existing work.

**Analogy and Language Models.** In cognitive science, analogy is formalized by *Structure-Mapping Theory*, which characterizes analogy as a mapping that preserves higher-order relations rather than surface features (Gentner & Markman, 1997; Gick & Holyoak, 1983). Building on this, in natural language processing, analogy has traditionally been studied through four-term lexical analogies ($A : B :: C : D$) (Turney et al., 2003; Mikolov et al., 2013a; Pennington et al., 2014; Ethayarajh et al., 2019; Ushio et al., 2021; Lewis & Mitchell, 2025; Musker et al., 2025). Recent work evaluates analogical reasoning as structured inference beyond lexical analogies, with benchmarks such as E-KAR and ANALOBENCH revealing persistent difficulties under increasing relational complexity (Chen, 2022; Ye, 2024). Our work complements this line of research by using synthetic tasks to precisely control relational structure and analyze how analogical reasoning emerges in Transformers.

**Understanding Transformers with Synthetic Tasks.** Transformer (Vaswani et al., 2017) has become the standard backbone of modern deep learning models, motivating interpretability studies (Bereska & Gavves, 2024). A prominent line of work uses synthetic tasks (Chan et al., 2022; Reddy, 2024; Nagarajan et al., 2025; Noroozizadeh et al., 2025) to isolate specific capabilities in controlled settings. Synthetic tasks have likewise been central to studying reasoning, including algorithmic and graph-structured problems (Zhang et al., 2025; Zhao et al., 2025; Qin et al., 2025) and compositional reasoning, where they have enabled analyses of training dynamics, representations, and scaling behavior (He et al., 2024; Wang et al., 2024; Furuta et al., 2024; Redhardt et al., 2025). We approach the line of work on understanding Transformers with synthetic tasks from the novel perspective of analogy, and show that the same nature extends to pretrained LLMs.

**Sample Efficiency and Creativity.** One of the major limitations of current LLMs is their poor sample efficiency (Warstadt et al., 2023): compared to humans, they require an enormous amount of data to acquire new knowledge. From a cognitive perspective, this remarkable sample

efficiency is often attributed, at least in part, to the use of analogy (Thagard, 1992; Gentner & Hoyos, 2017). Once humans learn relational structure in one domain, they can transfer it to a different but structurally similar domain, enabling rapid and highly data-efficient learning. Our results suggest that pretrained LLMs may also be capable of identifying shared relational structure via in-context learning (Figure 9 and Figure 10). However, our study does not yet establish whether such structure is effectively leveraged during learning to improve sample efficiency. Beyond sample efficiency, analogy has also been argued to play a central role in creativity (Goel, 1997; Gentner et al., 1997). Many historically important scientific discoveries are often described as arising from analogy or metaphor (Leatherdale, 1974; Winkler, 1981; Holyoak & Thagard, 1996). For example, Bohr's model of the atom inspired by planetary orbits (Bohr, 1913; Winkler, 1981). Scientific progress has often been driven by discovering structural similarities between seemingly *distant* domains. Capturing notions such as the "distance" between categories remains beyond the scope of our current task. Moreover, human reasoning operates over large collections of partially overlapping categories rather than cleanly disjoint ones, without any explicit functor signals ($\langle f \rangle$), a property that our synthetic setting does not yet model. Nevertheless, we view this work as an initial step toward mechanistically studying analogy in Transformers, and hope it will stimulate further research toward more sample-efficient and creative models. An extended discussion is deferred to Appendix M.

## 7. Limitations

Our controlled setup makes several simplifying assumptions. We also include additional ablations and analyses in the appendix that partially relax these assumptions.

**Explicit functor signal.** Our main synthetic task provides a dedicated functor token during training and evaluation. This is a convenient abstraction for isolating the mapping operation, but real-world analogies rarely come with such an explicit cue. As a first step toward relaxing this assumption, we consider an implicit-functor variant without an explicit functor token (Appendix Q), where analogical reasoning can still emerge, albeit in a more challenging regime.

**Perfect (or near-perfect) structural correspondence.** Our basic construction uses categories with (near-)isomorphic relational structure. Real-world analogies are often partial and noisy. To probe robustness beyond perfect isomorphism, we introduce controlled violations via a functor noise ratio and observe a consistent degradation in analogical performance as correspondence is broken (Appendix O).

**Complete relational graphs.** Our default construction uses complete relational graphs. In contrast, real-world relational structure is typically sparse and often exhibits small-world properties (Humphries & Gurney, 2008). We partially address this by studying sparse graphs in Appendix D.

**Restricted diversity of categories and mappings.** Our main experiments focus on two disjoint categories and a small number of mappings. In contrast, real-world reasoning often involves many partially overlapping categories with multiple competing correspondences. To test whether our findings are specific to a single toy mapping, we extend the setup to multiple categories and multiple functors and observe that analogical reasoning still emerges (Appendix P).

**External validity beyond synthetic tasks.** While synthetic tasks enable precise control and mechanistic analysis, they cannot fully capture the richness of natural language analogies. To partially address this gap, we evaluate on the natural-language E-KAR (Chen, 2022) benchmark and observe consistent layer-wise signatures linking Dirichlet energy reduction and improved answer probability (Appendix N).

**Limited causal validation of mechanistic claims.** Several of our mechanistic findings are supported primarily by correlational analyses. While we include additional ablations and complementary analyses in the appendix, stronger causal interventions—e.g., targeted activation patching (Wang et al., 2023) or editing of the relevant representation geometry (Wurgaft et al., 2026)—remain an important direction for future work.

## 8. Conclusion

We studied how analogical reasoning emerges in Transformers through a controlled synthetic task inspired by category-theoretic functors. Across data, optimization, and scaling sweeps, we found that analogical reasoning emerges later than memorization and compositional reasoning and is substantially more sensitive to training conditions. Through mechanistic analysis, we decomposed analogy into (i) geometric alignment of relational structure in the embedding space, quantified by Dirichlet Energy, and (ii) functor application implemented within the Transformer via attention and residual-stream composition. Finally, we observed closely related layer-wise signatures in pretrained LLMs, and further validated them on a natural-language analogy benchmark. We hope these results provide a concrete basis for future work on more naturalistic analogical reasoning settings and stronger causal tests of mechanistic hypotheses.

## Impact Statement

This work improves our understanding of analogical reasoning in Transformer-based models through controlled synthetic tasks. By linking insights from toy models to pretrained LLMs, it contributes to ongoing efforts in interpretability and model analysis. The work is foundational in nature and does not raise direct societal or safety concerns.

## Acknowledgements

We thank Heiga Zen for their support on this work and review on the initial version of the paper. We appreciate the funding support from Google Japan. This work was also supported by UTokyo-Google AI Symbiotic Future Society Program.

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

# A. Experiment Details

This appendix provides implementation details for the models and training procedures used in our synthetic experiments. All experiments were conducted on a single NVIDIA A100 GPU.

## A.1. Model Architecture

We use a lightweight causal Transformer model, similar to GPT-2 (Radford et al., 2019), augmented with Rotary Position Embeddings (RoPE) (Su et al., 2024), which is widely adopted in recent Transformer architectures. Unless otherwise specified, all experiments use the same architecture, summarized in Table 2.

*Table 2.* Transformer architecture used in synthetic experiments (GPT-2-like with RoPE).

| Component | Setting |
|---|---|
| Positional encoding | Rotary Position Embedding (RoPE) |
| Number of layers | 1 |
| Hidden size ($d_{\text{model}}$) | 128 |
| Number of heads | 1 |
| MLP width | $4 \times d_{\text{model}}$ |
| Dropout | 0.0 |
| Maximum sequence length | 64 |

## A.2. Optimization and Training

Models are trained using the Adam optimizer with standard hyperparameters. We apply a linear learning-rate warmup followed by a constant schedule. All default training hyperparameters are listed in Table 3.

*Table 3.* Training hyperparameters (default settings).

| Hyperparameter | Value |
|---|---|
| Optimizer | Adam |
| Learning rate | $1 \times 10^{-4}$ |
| Weight decay | 0.0 |
| Batch size | 64 |
| Number of epochs | 100 |
| Learning-rate schedule | Linear warmup then constant (warmup steps $= 0$) |
| Automatic mixed precision (AMP) | Enabled |

# B. Learning-rate sweeps for deeper models

In our main scaling experiments (Section 3.3), we used a fixed learning rate across model sizes. To test whether the observed underperformance of deeper models reflects an architectural limitation or suboptimal optimization settings, we perform a learning-rate sweep for deeper models. Figure 11 shows that with appropriate learning-rate choices, deeper models can achieve high analogical reasoning accuracy. This supports the interpretation that the depth effect is optimization-dependent, and further underscores the sensitivity of analogical reasoning to training hyperparameters.

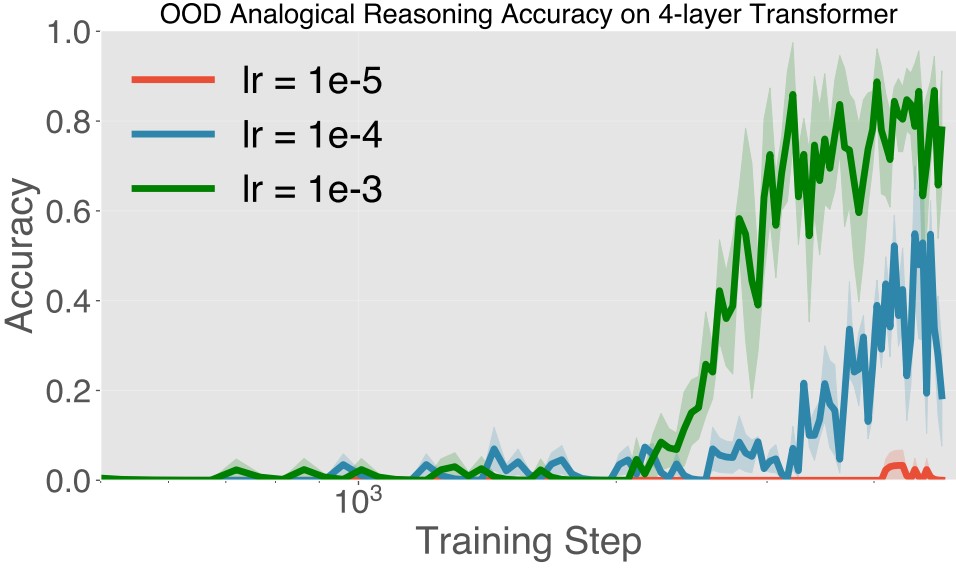

*Figure 11.* Learning-rate sweep for deeper models in the analogical reasoning task. With appropriate learning-rate choices, deeper models can achieve strong analogical reasoning performance, indicating that depth-related underperformance under default settings can be an optimization artifact.

# C. Transient Natures of Analogical Reasoning

A *transient nature* (Park et al., 2024; Singh et al., 2025) has been reported in in-context learning (ICL), where a capability that is once acquired can later be lost as training progresses. We observe a closely related phenomenon for analogical reasoning. Figure 12 shows the evolution of our internal mechanistic signals (Section 4) for the setting with $|\mathcal{R}| = 1,000$ relations, corresponding to the experiment described in Section 3.1. Although the model acquires analogical reasoning, the probability of predicting the correct target entity gradually decreases as training continues. Concurrently, the Dirichlet Energy increases, indicating a loss of geometric alignment in the embedding space. This suggests that the relational structure underlying analogical reasoning is no longer preserved. A similar phenomenon has been reported in prior work on in-context learning (Singh et al., 2025), which argues that circuits responsible for ICL can temporarily emerge during training, but later coexist and compete with alternative circuits (e.g., ICWL), eventually being suppressed as training continues. Consistent with this view, our results indicate that the structured embedding geometry supporting analogical reasoning can be transient: once the model begins to overly fit the training data, the previously acquired geometric alignment is disrupted. Notably, this behavior cannot be attributed to explicit regularization effects, as weight decay is set to zero in these experiments. Instead, our findings suggest that aggressive optimization toward training data fit can destabilize the geometric structures necessary for analogical reasoning, leading to its eventual degradation.

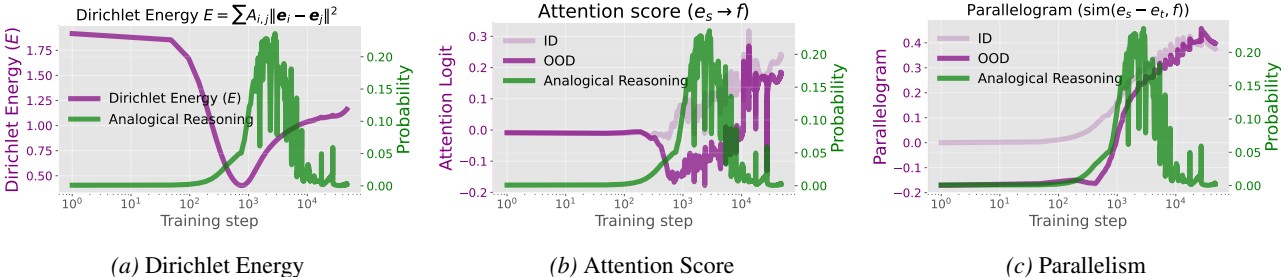

*(a)* Dirichlet Energy        *(b)* Attention Score        *(c)* Parallelism

*Figure 12.* Mechanistic signals underlying the emergence of analogical reasoning, where is measured by the model's probability of the correct target entity $(e_t)$. **(a) Dirichlet Energy** decreases during training, indicating increasing structural alignment in the embedding space. **(b) Attention Score** from the functor token $f$ to the source entity $e_s$ increase as analogical reasoning emerges, reflecting attention-based information retrieval. **(c) Parallelism**, defined by the similarity between $(e_t - e_s)$ and $f$, increases concurrently, indicating that the model realizes analogical reasoning by adding the functor representation $f$ to the source entity embedding $e_s$ via a residual connection.

## D. Effect of Graph Sparsity on Reasoning

In the main text, we assume that the relational graph within each category is complete, i.e., every pair of entities is connected by a relation. However, in real-world data, not all entity pairs are necessarily related. Many real-world relational structures are known to be sparse and often exhibit small-world properties, with dense local connectivity but missing edges globally (Humphries & Gurney, 2008). Figure 13-(a) illustrates a comparison between a complete graph and a non-complete (sparse) graph. To study the effect of graph completeness, we analyze how the sparsity of the relational graph influences compositional and analogical reasoning. Specifically, we remove a fraction of atomic facts from the training data, where each atomic fact corresponds to a triple $(e_s, r, e_t)$. This removal effectively increases the sparsity of the underlying relational graph.

Figure 13-(b) shows the compositional and analogical reasoning performance as a function of the removed atomic fact ratio. We observe that compositional reasoning remains robust even under substantial sparsity. In contrast, analogical reasoning fails to emerge as the graph becomes increasingly sparse, suggesting that analogical reasoning critically relies on sufficiently dense relational structure.

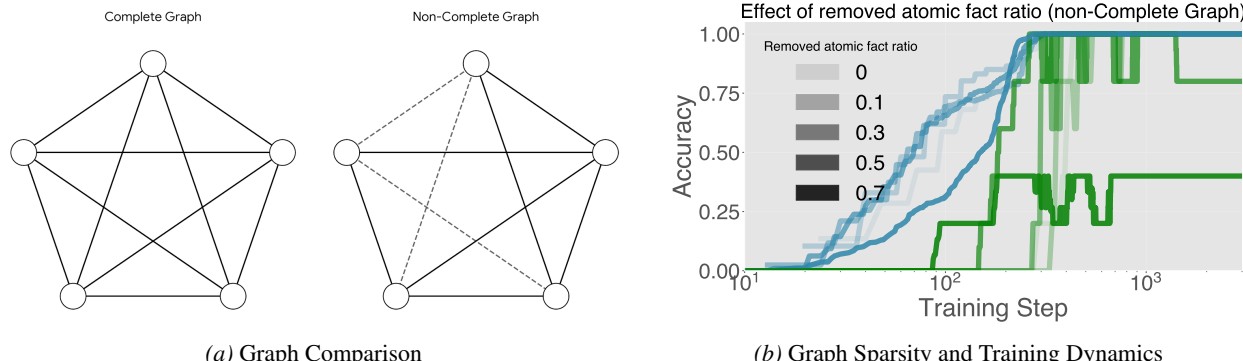

*(a)* Graph Comparison        *(b)* Graph Sparsity and Training Dynamics

*Figure 13.* Effect of graph sparsity on compositional and analogical reasoning. (a) Comparison between a complete relational graph and a non-complete (sparse) graph. (b) Composition reasoning and analogical reasoning performance, which controls the sparsity of the relational graph. While compositional reasoning remains robust to sparsity, analogical reasoning degrades and eventually fails as the graph becomes increasingly sparse.

## E. Effect of Learning Rate

We investigate the effect of the learning rate on the acquisition of compositional and analogical reasoning. As shown in Figure 14, when the learning rate is set too high, the model fails to reliably acquire analogical reasoning, and even compositional reasoning becomes difficult to learn. In contrast, smaller learning rates allow the model to first fit the training data and subsequently exhibit generalization behavior. This observation is consistent with prior findings in the grokking literature, which show that excessively large learning rates can prevent the emergence of generalization phenomena (Liu et al., 2022). Our results suggest that the emergence of analogical reasoning is similarly sensitive to optimization dynamics, and can be hindered by overly aggressive learning rates.

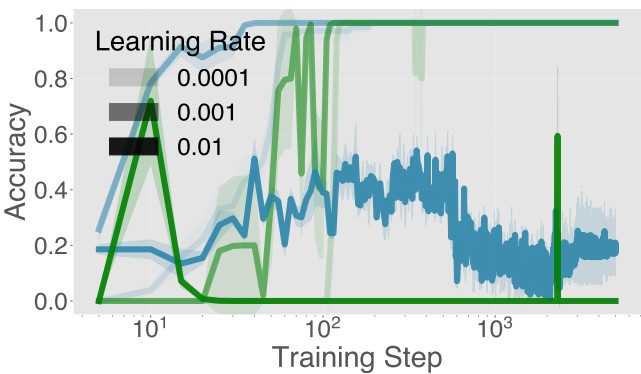

*Figure 14.* Effect of learning rate on out-of-distribution (OOD) performance. Large learning rates hinder the acquisition of both compositional and analogical reasoning, while smaller learning rates enable gradual generalization.

# F. Dynamics of PCA visualizations

We provide a qualitative visualization of how entity embeddings evolve during training. We apply Principal Component Analysis (PCA) to the embedding vectors at different training epochs and project them onto a two-dimensional space. This allows us to track the temporal dynamics of representation geometry throughout optimization.

As shown in Figure 15, embeddings at early training stages exhibit little discernible structure and largely overlap in the projected space. As training proceeds, a more coherent geometric organization gradually emerges, with entities becoming arranged according to their underlying relational roles. This observation suggests that structural organization in the embedding space is not present a priori, but is progressively formed through learning.

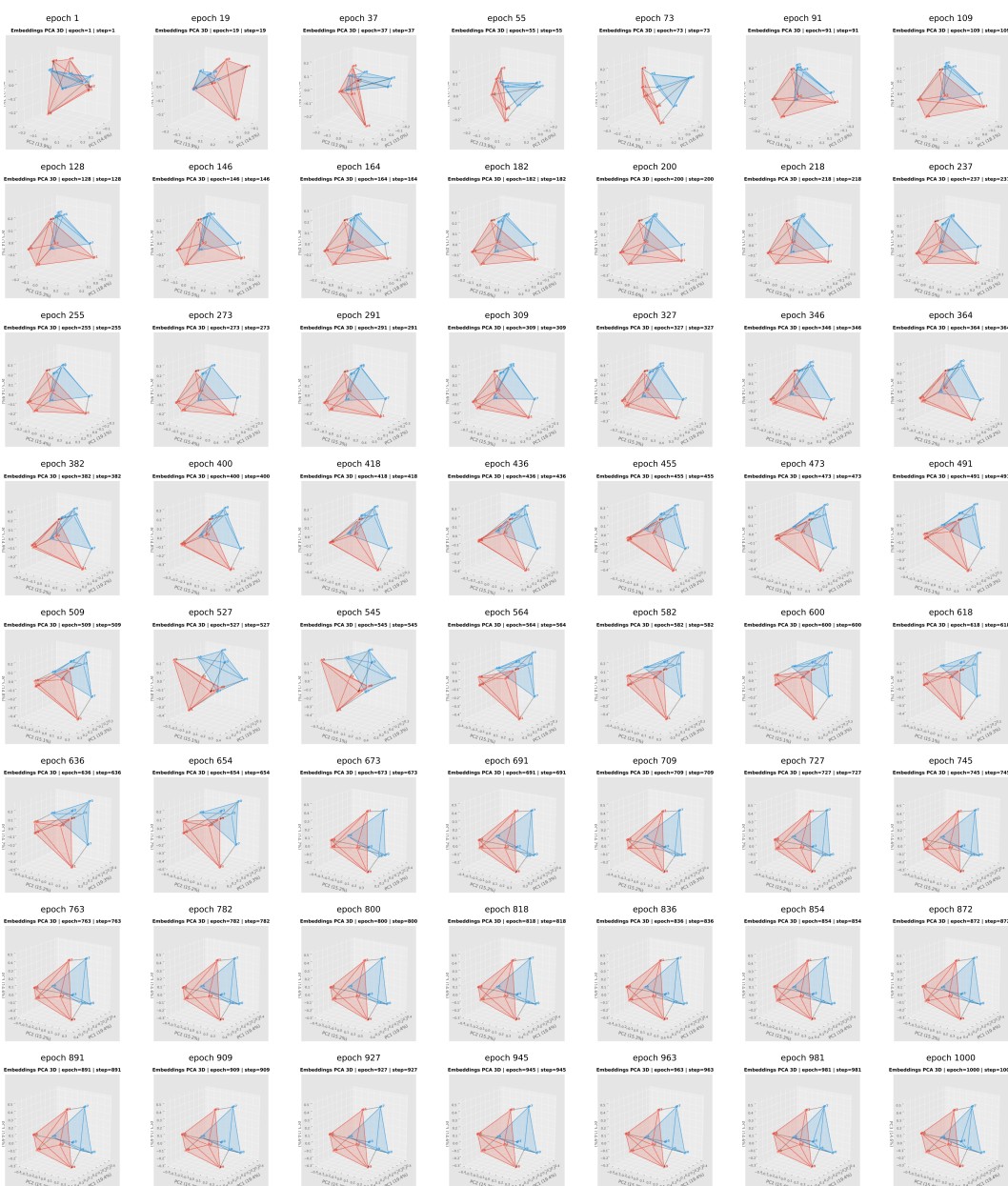

*Figure 15.* Dynamics of PCA visualizations of entity embeddings throughout training. Each panel shows the projection of embeddings at a different training epoch. In early stages, embeddings are largely unstructured and overlapping. As training progresses, coherent geometric structure gradually emerges, with entities organizing according to their underlying relational roles. This illustrates that structural alignment in representation space is not present initially but forms progressively during optimization.

# G. Derivation of Multi-dimensional Dirichlet Energy

Following (Park et al., 2025), we provide a detailed derivation of the Dirichlet energy for multi-dimensional node representations, which is used throughout our analysis to quantify structural alignment in representation space.

**Scalar-valued signal.**  Let $\mathcal{G} = (\mathcal{V}, \mathcal{E})$ be an undirected graph with $n = |\mathcal{V}|$ nodes. Let $\boldsymbol{A} \in \mathbb{R}^{n \times n}$ denote its (possibly weighted) adjacency matrix, and let $\boldsymbol{x} \in \mathbb{R}^n$ be a scalar signal defined on the nodes, where $\boldsymbol{x}_i$ denotes the value associated with node $i$. The Dirichlet energy of $\boldsymbol{x}$ on graph $\mathcal{G}$ is defined as

$$E_{\mathcal{G}}(\boldsymbol{x}) = \sum_{i,j} \boldsymbol{A}_{i,j} (\boldsymbol{x}_i - \boldsymbol{x}_j)^2. \tag{4}$$

This quantity measures the smoothness of the signal with respect to the graph structure: neighboring nodes incur a high energy penalty if their assigned values differ significantly.

**Multi-dimensional signal.**  We now extend this definition to the case of multi-dimensional node representations. Let $\boldsymbol{X} \in \mathbb{R}^{n \times d}$ be a matrix of node embeddings, where each node $i$ is associated with a vector $\boldsymbol{x}_i \in \mathbb{R}^d$, and $\boldsymbol{x}_{i,k}$ denotes its $k$-th component. A natural extension of the Dirichlet energy is obtained by summing the scalar Dirichlet energy over each dimension:

$$E_{\mathcal{G}}(\boldsymbol{X}) = \sum_{k=1}^{d} \sum_{i,j} \boldsymbol{A}_{i,j} (\boldsymbol{x}_{i,k} - \boldsymbol{x}_{j,k})^2. \tag{5}$$

Rearranging the summation, this can be written equivalently as

$$E_{\mathcal{G}}(X) = \sum_{i,j} \boldsymbol{A}_{i,j} \sum_{k=1}^{d} (\boldsymbol{x}_{i,k} - \boldsymbol{x}_{j,k})^2 \tag{6}$$

$$= \sum_{i,j} \boldsymbol{A}_{i,j} \|\boldsymbol{x}_i - \boldsymbol{x}_j\|_2^2, \tag{7}$$

where $\|\cdot\|_2$ denotes the Euclidean norm. Thus, the multi-dimensional Dirichlet energy penalizes large pairwise distances between representations of adjacent nodes in the graph.

# H. Embedding Structure under Model Scaling

Figure 16 visualizes the entity embedding structure after training for models with different depths. While a 1-layer Transformer exhibits clear geometric alignment between the two categories, this alignment is largely absent in the 4-layer model.

This observation supports the view that analogical reasoning is not primarily determined by model capacity, but rather by whether the model discovers the aligned geometric structure in the embedding space. Moreover, increasing the number of parameters expands the space of solutions that fit the training data, and can promote memorization or locally sufficient strategies that do not enforce such global alignment. As a result, larger or deeper models may achieve low training loss without forming the structured representations required for analogy.

We emphasize that this effect depends on the optimization setting and data regime used in our experiments. Different regularization schemes or training objectives may bias larger models toward more geometrically aligned solutions. Nevertheless, under our setup, increased model capacity alone does not reliably induce the embedding structure necessary for analogical reasoning.

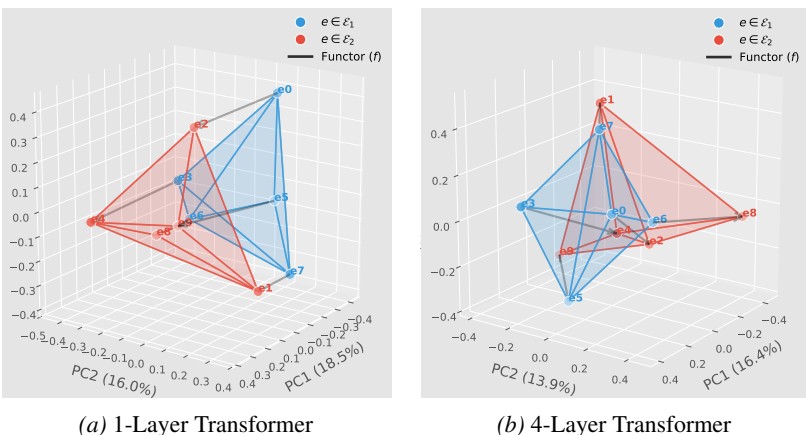

*(a)* 1-Layer Transformer        *(b)* 4-Layer Transformer

*Figure 16.* PCA Visualization of entity embeddings after ($10^3$ step) the acquisition of analogical reasoning. Entity embeddings from category $\mathcal{E}_1$ (blue) and $\mathcal{E}_2$ (red) are shown, with arrows indicating the functor. **(a) 1-Layer Transformer**, embeddings from the two categories are structurally aligned. **(b) 4-Layer Transformer**, embeddings from the two categories are not structurally aligned.

# I. Output Probability under the Alternative Prompt Designs

In this section, we report how different prompt designs affect the next-token prediction probabilities of GEMMA2-2B. Specifically, we show the top-5 output probabilities for the next token under several prompt variants below.

**Prompt 1** corresponds to the prompt used throughout our main experiments (see Figure 8). The correct answer for this prompt is token 7, and the model assigns a high probability to this token.

> Prompt 1 (Target entity is `<e7>`)
>
> ```
> <e1>a<e2>, <e1>b<e3>.  <e6>a<e4>, <e6>b<e7>.  <e1>~<e6>, <e3>~<e
> ```

**Prompt 2** is a naive variant that uses the categories Category 1 ($\{e_1, e_2, e_3\}$) and Category 2 ($\{e_4, e_5, e_6\}$). The correct answer in this case is token 6, and the model predicts this token with high probability. However, this prompt introduces an explicit arithmetic correspondence between Category 1 ($\{e_1, e_2, e_3\}$) and Category 2 ($\{e_4, e_5, e_6\}$), which can be interpreted as a fixed "+3" mapping. As a result, the task can be solved without genuine analogical reasoning. For this reason, we do not use this prompt in our main experiments.

> Prompt 2 (Target entity is `<e6>`)
>
> ```
> <e1>a<e2>, <e1>b<e3>.  <e4>a<e5>, <e4>b<e6>.  <e1>~<e4>, <e3>~<e
> ```

**Prompt 3** follows the same structural pattern as Prompt 1 and uses exactly the same tokenization scheme as our synthetic task Table 1, but the model fails to reliably predict the correct answer (token 7). We hypothesize that this failure is due to unintended priors introduced during pretraining. In particular, the symbols $e$, $r$, and $f$ may carry semantic or syntactic biases from pretraining that interfere with analogical reasoning. Moreover, tokens such as `<r12>` are split into multiple sub-tokens (e.g., `<, r, 1, 2, >`), which may make it more difficult for the model to capture the intended relational structure.

> Prompt 3 (Target entity is `<e7>`)
>
> ```
> <e1><r12><e2>, <e1><r23><e3>.  <e6><r12><e4>, <e6><r23><e7>.  <e1><f><e6>, <e3><f><e
> ```

**Prompt 4** is a simplified variant of Prompt 1 in which special markers such as `<` and `e` are removed. Under this prompt, the model fails to produce the correct answer. This suggests that explicit entity markers (`<`, `>`) play an important role in helping the model recognize and track entities, and their removal significantly degrades performance.

> Prompt 4 (Target entity is `7`)
>
> ```
> 1a2, 1b3.  6a4, 6b7.  1~6, 3~
> ```

| **Prompt 1** | | | **Prompt 2** | | | **Prompt 3** | | | **Prompt 4** | | |
|---|---|---|---|---|---|---|---|---|---|---|---|
| # | Token | Prob | # | Token | Prob | # | Token | Prob | # | Token | Prob |
| 1 | 7 | 0.74 | 1 | 6 | 0.71 | 1 | 6 | 0.41 | 1 | 6 | 0.46 |
| 2 | 4 | 0.08 | 2 | 5 | 0.12 | 2 | 7 | 0.27 | 2 | 5 | 0.11 |
| 3 | 6 | 0.07 | 3 | 4 | 0.09 | 3 | 4 | 0.09 | 3 | 7 | 0.09 |
| 4 | 5 | 0.03 | 4 | 2 | 0.04 | 4 | 1 | 0.08 | 4 | 4 | 0.08 |
| 5 | 2 | 0.03 | 5 | 2 | 0.02 | 5 | 2 | 0.05 | 5 | 8 | 0.08 |

*Figure 17.* Output probability under several prompt variants. The colored row denotes the correct answer.

## J. Impact of the Number of Entities

We analyze how the number of entities affects analogical reasoning in LLMs, and its relationship with Dirichlet Energy. Example prompts used in this analysis are shown below.

---

Number of Entities is 4 (Target is `<e8>`)

```
<e1>a<e2>, <e1>b<e3>, <e1>c<e4>.   <e6>a<e5>, <e6>b<e7>, <e6>c<e8>.   <e1>˜<e6>,
<e2>˜<e5>, <e3>˜<e7>, <e4>˜<e
```

---

Number of Entities is 5 (Target is `<e10>`)

```
<e1>a<e2>, <e1>b<e3>, <e1>c<e4>, <e1>d<e5>.   <e7>a<e6>, <e7>b<e8>, <e7>c<e9>,
<e7>d<e10>.   <e1>˜<e7>, <e2>˜<e6>, <e3>˜<e8>, <e4>˜<e9>, <e5>˜<e
```

---

Number of Entities is 7 (Target is `<e14>`)

```
<e1>a<e2>, <e1>b<e3>, <e1>c<e4>, <e1>d<e5>, <e1>e<e6>, <e1>f<e7>.   <e9>a<e8>,
<e9>b<e10>, <e9>c<e11>, <e9>d<e12>, <e9>e<e13>, <e9>f<e14>.   <e1>˜<e9>, <e2>˜<e8>,
<e3>˜<e10>, <e4>˜<e11>, <e5>˜<e12>, <e6>˜<e13>, <e7>˜<e
```

---

As shown in Figure 18, the relationship between analogical reasoning performance and the decrease of Dirichlet Energy is consistently observed across all prompts. Notably, as the number of entities increases, the layer at which analogical reasoning performance peaks shifts toward later layers. This trend is consistent with our findings in the toy task Section 3.1, and suggests that more complex analogical problems require deeper computation to align relational structure.

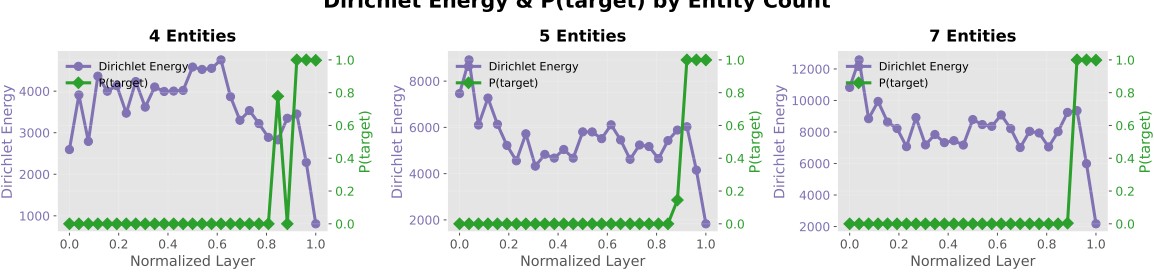

*Figure 18.* Effect of the number of entities on analogical reasoning in LLMs. Across different entity counts, analogical reasoning performance exhibits a consistent relationship with Dirichlet energy. As the number of entities increases, the emergence of analogical reasoning shifts toward later layers, indicating increased computational depth.

## K. Dirichlet Energy in Llama

We observe the same qualitative relationship between analogical reasoning performance and Dirichlet Energy in LLaMA. As shown in Figure 19, increasing the number of entities leads to similar layer-wise trends as in other LLMs, indicating that the geometric mechanism underlying analogical reasoning generalizes across model families.

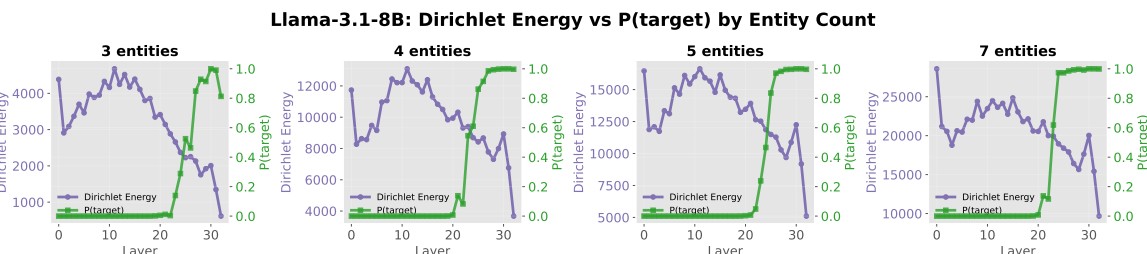

*Figure 19.* Relationship between analogical reasoning performance and Dirichlet energy in LLaMA across different numbers of entities. The same qualitative trends observed in other LLMs persist, indicating that the underlying geometric mechanism is not model-specific.

# L. PCA of LLM Hidden States in All Layers

We analyze the internal representations of pretrained large language models (LLMs) when they are prompted to perform analogical reasoning. Specifically, we apply PCA to the hidden states at each Transformer layer and visualize how the geometry of entity representations evolves across depth.

As shown in Figure 20, representations in earlier layers exhibit limited geometric organization across categories. However, as depth increases, entities belonging to corresponding categories become progressively aligned in the representation space. In later layers, this alignment becomes particularly pronounced, indicating that analogical structure is increasingly encoded as the model approaches the output layers.

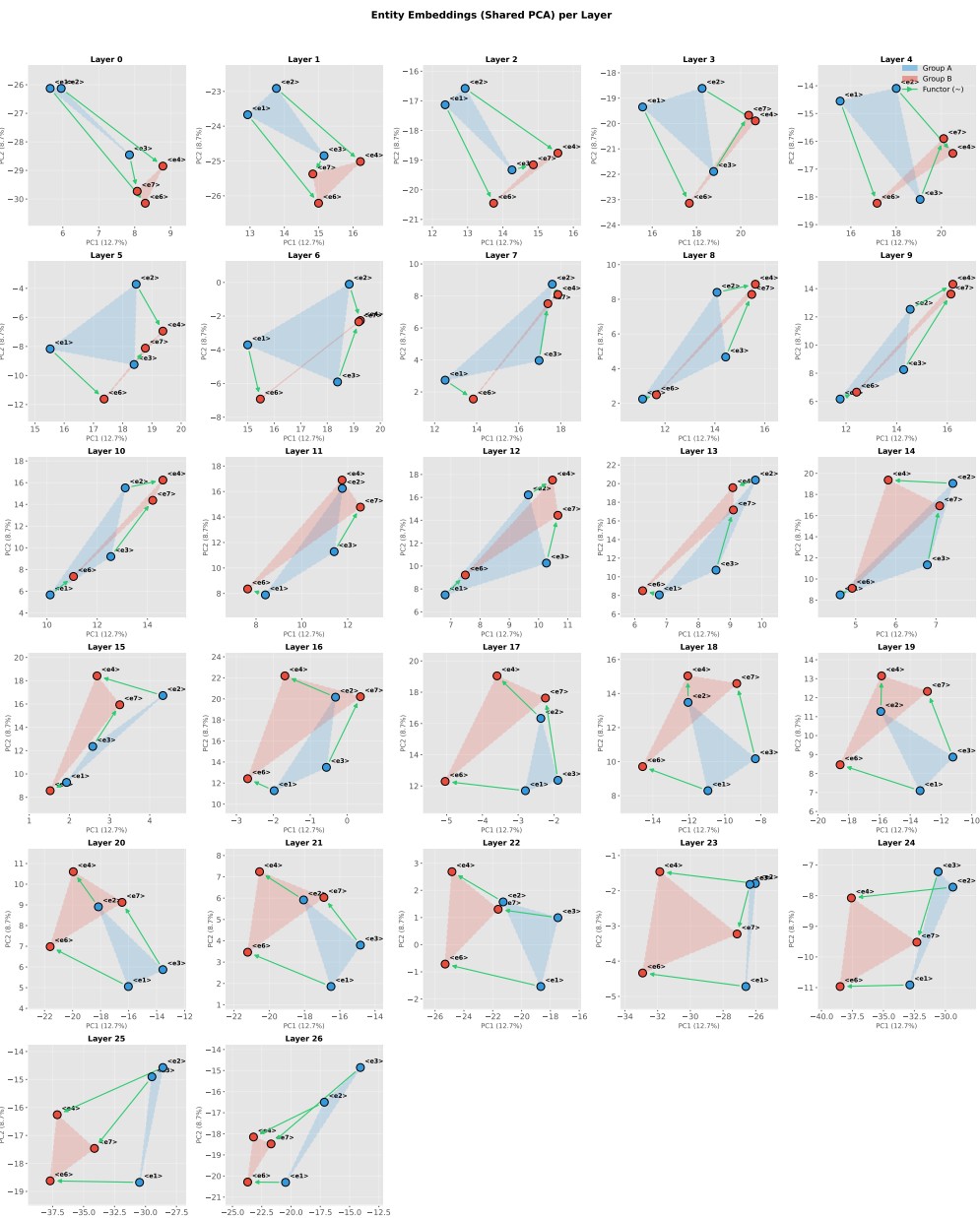

*Figure 20.* PCA visualizations of LLM hidden states across Transformer layers when prompted with an analogical reasoning task. As depth increases, entity representations from corresponding categories become geometrically aligned, indicating the progressive emergence of analogical structure in later layers.

# M. More Discussions

**Linear Representation Hypothesis.** In the context of language models, it has long been argued that high-level semantic concepts are represented linearly in the embedding space, a view commonly referred to as the *Linear Representation Hypothesis* (Mikolov et al., 2013b; Pennington et al., 2014; Park et al., 2023). A canonical example is the observation that vector differences such as woman − man ≈ queen − king capture semantic relations through linear offsets. Beyond lexical relations, prior work has suggested that more abstract transformations, such as mappings between languages (e.g., English → French), may also be encoded as approximately linear directions in representation space (Park et al., 2023). This perspective aligns with the mechanism we describe in Section 4, where such linear transformations can be interpreted as functorial mappings that enable conceptual leaps across categories. Previous studies have also shown that such subspace structures can be acquired along the depth of a Transformer through in-context learning (Hendel et al., 2023; Cho et al., 2025). These works do not explicitly connect linear representations to analogical reasoning. In particular, it remains unclear how models learn to identify entities that play the same relational role across distinct domains, a core requirement for analogy. Our work bridges this gap by explicitly linking linear structure in representation space to analogical reasoning: we show that analogy emerges when functor-like transformations become geometrically aligned across categories, enabling the model to infer correspondences between role-equivalent entities in different domains.

**A Category-Theoretic Perspective** In category theory (Awodey, 2010), a category consists of *objects* and *arrows* (or morphisms) between them. A key insight of category theory is that neither objects nor arrows possess intrinsic meaning; they are abstract symbols defined purely by their relationships to one another. This perspective closely mirrors the learning setting of language models. A language model operates on sequences of token IDs and is trained to predict the next token. It has no access to the intrinsic semantics of tokens, and instead learns entirely from the relationships among symbols. In this sense, meaning emerges from relational structure rather than being predefined.

In our task, analogical reasoning corresponds to identifying similarities between relational structures that emerge from such interactions. The *functor* in our formulation captures this notion: it represents a mapping between relational structures, rather than between individual symbols themselves.

# N. E-KAR: Dirichlet energy and answer probability across layers

To test whether the geometric alignment signature extends beyond our synthetic task, we analyze Gemma-2-9B on the E-KAR natural-language analogy benchmark (Chen, 2022). E-KAR is a multiple-choice analogy dataset (868 examples) where each question asks the model to identify a candidate pair that shares the same relation as a given source pair.

**Setup.** For each question, we consider the hidden states at the entity-pair positions used to represent the source relation and each candidate relation. We compute the Dirichlet energy between the source and the correct candidate pair across layers, and track the model's probability assigned to the correct option using logit-lens decoding.

**Result.** As shown in Figure 21, Dirichlet energy decreases across depth, and this decrease is followed by an increase in $p(\text{target})$ in later layers. This mirrors the synthetic-task observation that geometric alignment (lower energy) co-occurs with improved analogical prediction.

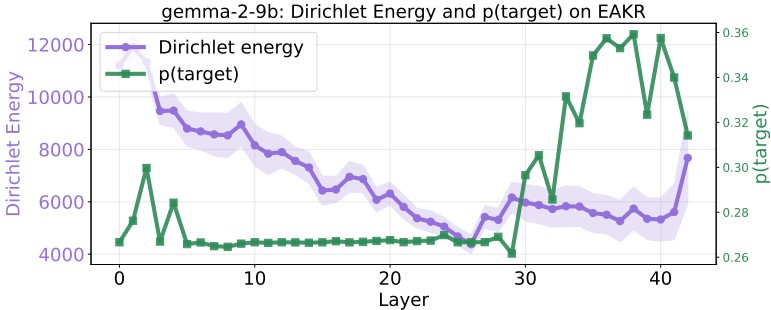

*Figure 21.* Gemma-2-9B: Dirichlet energy and $p(\text{target})$ across layers on E-KAR (Chen, 2022).

# O. Imperfect isomorphism: functor noise-ratio ablation

Our main synthetic setup assumes perfectly isomorphic relational structure across categories. To relax this assumption, we introduce a *noise ratio* when transferring edges across categories via the functor. For a fraction of transferred edges, we randomly replace the original relation label with another relation sampled from $\mathcal{R}$, thereby breaking exact structural correspondence.

**Result.** Figure 22 shows that analogical reasoning degrades consistently as the noise ratio increases. This provides a controlled demonstration that analogical generalization becomes harder under partial/noisy cross-domain correspondence.

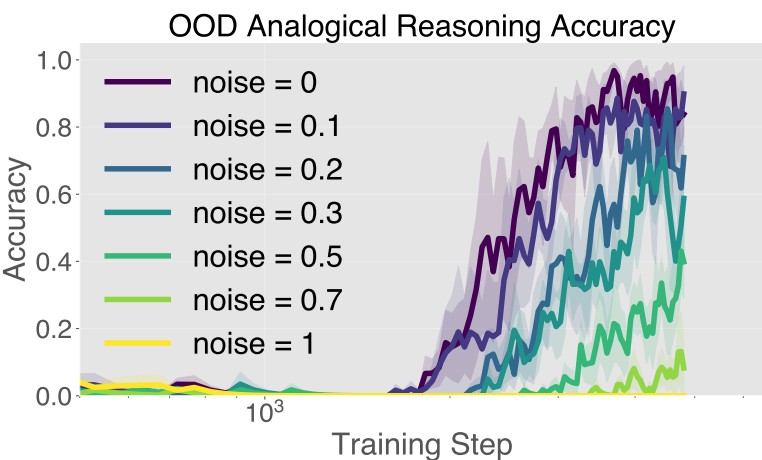

*Figure 22.* OOD analogical reasoning accuracy under imperfect isomorphism (functor noise ratio).

# P. Multiple functors/categories

A potential concern is that our main experiments involve a single functor token and a single cross-category mapping. To test whether the phenomenon persists with multiple mappings, we extend the setup to multiple categories with multiple functor tokens.

**Setup.** We fix the total number of entities to 100 and vary the number of categories: 2 categories ($50 \times 2$; the main-text setting), 4 categories ($25 \times 4$), and 5 categories ($20 \times 5$). For each non-base category, we introduce a distinct functor token (e.g., $f_1$, $f_2$, $f_3$) mapping from the base category to the target category (schematic in Figure 23, left).

**Result.** As shown in Figure 23 (right), analogical reasoning consistently emerges across all settings. Notably, emergence is fastest in the 5-category case, which we attribute to the smaller number of entities per category (consistent with the entity-count trends in Section 3.1).

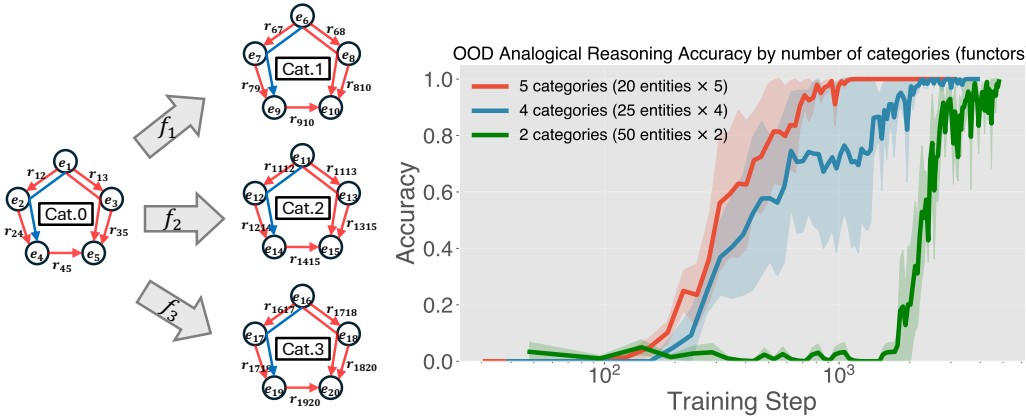

*Figure 23.* Multiple-functor setting: schematic (left) and OOD analogical reasoning accuracy by number of categories (right).

## Q. Implicit functor variant (no explicit functor token)

Real-world analogies rarely come with an explicit mapping cue. To test whether analogical reasoning can emerge without an explicit functor token, we consider a variant in which the model receives only entity-relation facts, and the mapping signal is implicit.

**Setup.** We compare our original formulation (explicit functor query) against a no-functor formulation in which the query provides *no* functor token and instead specifies two partial tuples of entities, one from the source category and one from the target category. Concretely, the input has the form `<e1><e2><e3><e6><e7>` (and the model is trained/evaluated to predict the next token `<e8>`). Intuitively, the model must infer the latent correspondence from relational roles (i.e., the fact that `<e1>,<e2>,<e3>` and `<e6>,<e7>,<e8>` should form structurally matching tuples) without being given an explicit mapping cue.

**Result.** Figure 24 shows that analogical reasoning can still emerge in the implicit setting, although it is more challenging and tends to saturate at a lower accuracy.

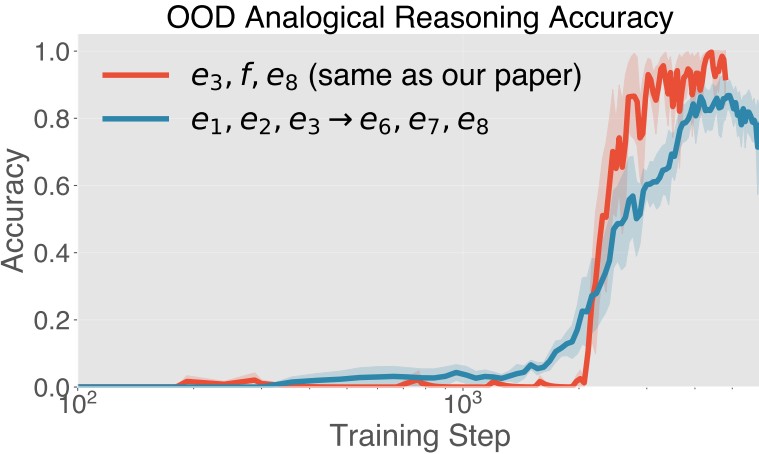

*Figure 24.* OOD analogical reasoning accuracy with and without an explicit functor token.

# R. Logit lens vs. linear probe

Logit lens is a convenient tool to track intermediate "beliefs," but it can underestimate information encoded in early-layer representations due to mismatch with the final unembedding. To address this, we compare logit-lens decoding with a learned linear probe.

**Setup.** Using Gemma-2-9B, we train a linear classifier on per-layer hidden states to predict the correct target option for the analogy questions. We then track the probe's probability for the correct answer across layers, alongside Dirichlet energy.

**Result.** As shown in Figure 25, both readouts improve as Dirichlet energy decreases, with the linear probe typically rising earlier than logit lens. This supports the interpretation that the layer-wise reduction in Dirichlet energy reflects increasingly linearly decodable analogical structure.

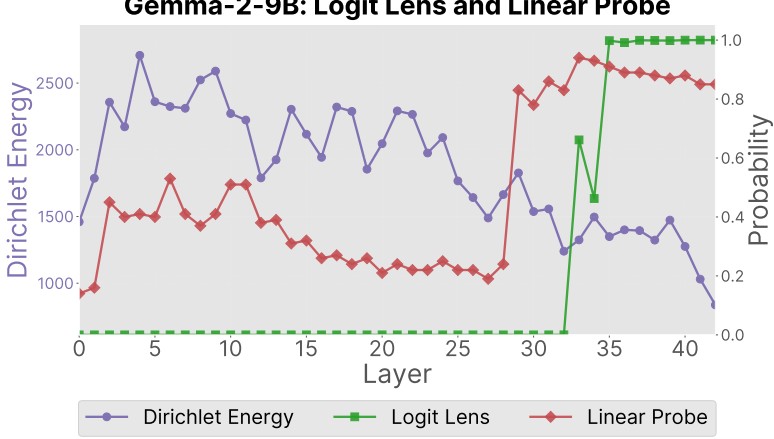

*Figure 25.* Gemma-2-9B: Dirichlet energy with logit-lens and linear-probe probabilities across layers.

# S. Dirichlet energy and embedding norm during training

Dirichlet energy depends on representation distances. In cross-entropy training, embedding norms can continue to grow even after loss/accuracy saturates, which can cause energy to increase slightly despite stable task performance. Figure 26 illustrates this behavior by jointly plotting energy and the embedding $\ell_2$ norm over training steps.

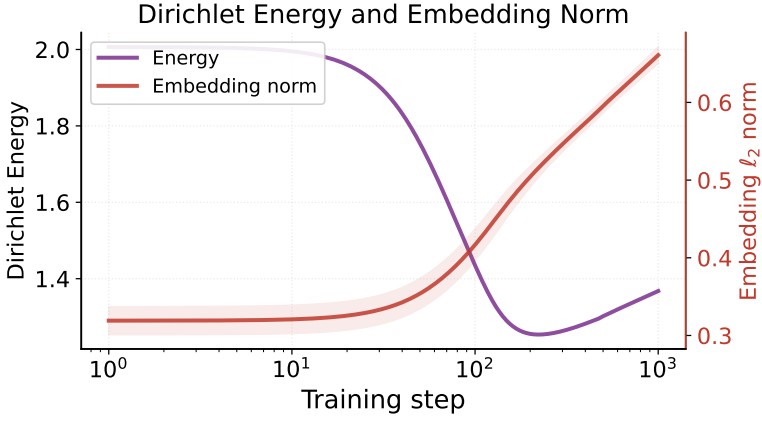

*Figure 26.* Dirichlet energy and embedding norm over training steps.

## T. Bilinear probe analysis: functor direction and relational interaction

While our main mechanistic claim emphasizes an additive functor direction, alternative relational parameterizations are possible. Following a bilinear view of relations, we train a bilinear probe that predicts whether a pair of entities $(e_i, e_j)$ is in a specific relation by a score $e_i^\top W e_j$.

**Setup.** We train the bilinear probe on the learned entity embeddings and achieve 81.1% accuracy. We then quantify the interaction between the learned relation matrix $W$ and the functor direction $f$ by tracking a "null-space ratio" during training.

**Result.** As shown in Figure 27, the null-space ratio decreases over training (from 0.624 to 0.108), suggesting that $f$ becomes a direction with minimal relational interaction under the bilinear formulation. This provides complementary geometric evidence consistent with the functor-direction interpretation.

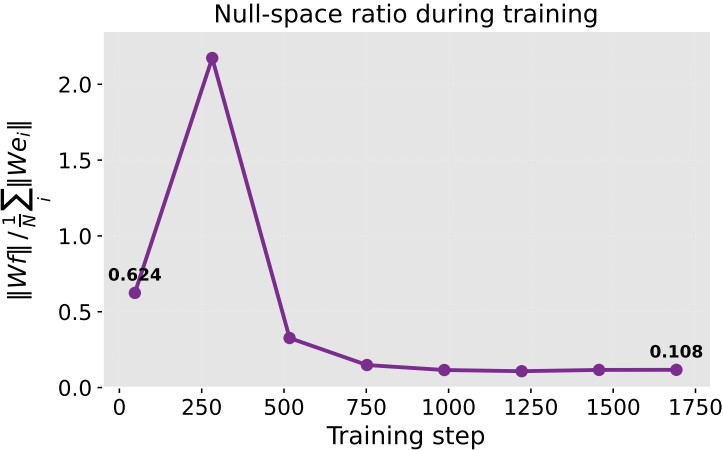

*Figure 27.* Null-space ratio during training (bilinear probe analysis).

