# OpenReview forum: "Emergent Analogical Reasoning in Transformers"
_ICML.cc/2026/Conference — ICML 2026 spotlight_

### Official Review · Reviewer_QSCk · 2026-03-07

**Soundness:** 4
**Presentation:** 4
**Significance:** 3
**Originality:** 4
**Overall Recommendation:** 6
**Confidence:** 4

**Summary:**

This paper studies how and why analogical reasoning emerges in transformer models (e.g. the sun in the solar system is like a proton in an atomic structure). Inspired by category theory, the authors formalize analogy as functor-based inference of cross-category correspondence. They design a synthetic task that evaluates analogical reasoning as well as the widely studied compositional reasoning (A-->B and B-->C, therefore A-->C). They train a 1-layer transformer, and show that memorization happens first, then compositional generalization and then analogical reasoning. The authors then identify a mechanism underlying analogical reasoning: (1) geometric alignment of entity embeddings across categories, which they quantify via Dirichlet energy and (2) functor application as vector addition within the model. They show that these mechanistic signatures also apply in trained LLMs.

**Compliance With Llm Reviewing Policy:**

Affirmed.

**Final Justification:**

This paper makes a genuine and well-executed contribution to mechanistic interpretability by formalizing analogical reasoning as functor-based inference and identifying the computational mechanisms underlying its emergence in transformers. The category theory framing is principled and productive, the three-stage learning dynamics are clearly demonstrated, and the connection to pretrained LLMs is a meaningful empirical contribution.

My original concerns were: (1) the reliance on a synthetic setting with strong assumptions (perfect isomorphism, explicit functor token); (2) the inverse scaling result with depth potentially undermining generality; and (3) the causal claim about geometric alignment resting on correlational evidence.

The rebuttal addressed all three convincingly with novel experiments. The noise-ratio ablation shows clean performance degradation under imperfect isomorphism. The learning rate sweep demonstrates that inverse scaling with depth is an optimization artifact rather than a fundamental limitation. Most importantly, the layer-skipping causal intervention provides direct causal evidence for the geometric alignment hypothesis. The E-KAR experiments further strengthen generalization to naturalistic analogy tasks.

The rebuttal substantially improved my assessment and I am upgrading my score to Strong Accept. As a final note, the implicit functor experiment is promising and I would encourage the authors to develop it further in the revision.

**Key Questions For Authors:**

1.	The functor token f is explicitly provided during training and evaluation. Have the authors tried to run a variant of their experiment where this signal is absent or implicit?
2.	Similarly, have the authors tried to break the perfect isomorphism? How does that affect analogical reasoning and the geometric representations?
3.	What does the inverse scaling result (deeper models don’t show geometric alignment) mean for the generality of the mechanistic theory?

**Limitations:**

Yes

**Strengths And Weaknesses:**

**Soundness**

*Strengths*

-	The category theory framing is well-motivated and useful, and formalizing analogy as a functor is a principled choice which clearly separates analogical from compositional reasoning
-	The mechanistic analysis is well motivated and backed by multiple metrics.
-	The fact that these metrics also apply to LLMs is a strong empirical contribution.

*Weaknesses*

-	The synthetic setting rests on some strong assumptions: two disjoint categories with perfectly isomorphic relational structure and an explicit functor token <f>. That last assumption is particularly limiting, since human analogical reasoning typically requires inferring analogies without a specific signal. The authors acknowledge these limitations, but only in Appendix L. This deserves more prominent treatment in the main text or, ideally, an experiment testing how reliant the findings are on these assumptions. The authors already tested for the effect of graph sparsity in Appendix C, so a test of the effect of non-perfect isomorphism should not be hard to run
-	The inverse scaling result, where deeper models do worse at analogy and show less geometric alignment (appendix G), seems to somewhat undermine the generality of the findings. Why do deeper models show less alignment and what does this mean for modern LLMs which are much deeper than the ones tested here?
-	The core causal claim (geometric alignment in embedding space enables analogical reasoning) is supported by correlational evidence in LLMs. A more direct test would be to intervene on the embedding geometry.

**Presentation**

*Strengths*

-	The paper is clearly written and well-organized.
-	The figures are well illustrated and easy to understand.

**Significance**

*Strengths*

-	Analogy is central to human intelligence and creativity, but mechanistic accounts of how this work are rare. This paper makes an important step towards filling that gap.
-	The empirical findings, like the three-stage learning dynamics and the effects of data dependence, are interesting and have potential implications for training design.

*Weaknesses*

-	It’s still unclear how well this synthetic setting applies to real-world analogy. While the authors show signatures of the identified mechanism in small LLMs, the scaling effects they found suggest that these might disappear in larger models.

**Originality**

*Strengths*

-	Connecting functors from category theory to analogical reasoning in transformers is, to my knowledge, novel and a worthy contribution.
-	The observation that LLMs implement analogy via layer-wise geometric alignment is a new and interesting perspective

---

> ### Author Rebuttal · Authors · 2026-03-31
>
> We appreciate the reviewer’s careful reading and valuable feedback.
>
> ---
>
> **>W1&Q2**
>
> >  a test of the effect of non-perfect isomorphism should not be hard to run
>
> > have the authors tried to break the perfect isomorphism?
>
> As noted in **Appendix L (Limitations)**, perfect isomorphism is a simplification, while real-world analogies are often partial and noisy.
>
> To address this, we conducted additional experiments where we break the perfect isomorphism. In our original setup, relations are transferred exactly across categories via the functor (i.e., $(e_s, r, e_t) \rightarrow (\mathcal{F}(e_s), r, \mathcal{F}(e_t))$), resulting in identical relational structures.
>
> In the new setting, we relax this constraint. We introduce a **noise ratio** parameter: when transferring edges via the functor, we randomly replace the relation $r$ with a different relation for a certain fraction of edges, thereby perturbing the structural correspondence between categories.
>
> The results (link: https://anonymous.4open.science/r/Emergent_Analogy_rebuttal-C3E9/functor_noise_ratio_acc.pdf) show that as the noise ratio increases, analogical reasoning performance degrades consistently.
>
> This setting provides a more realistic scenario, and we will include these results in the appendix.
>
> ---
>
> **>W2&Q3**
> > The inverse scaling result,
>
> > What does the inverse scaling result
>
> We agree that the observed inverse scaling with depth may appear to undermine the generality of our findings. During the rebuttal period, we conducted additional experiments and found that larger/deeper models can achieve strong performance when the optimizer is properly tuned.
>
> In additional experiments (link: https://anonymous.4open.science/r/Emergent_Analogy_rebuttal-C3E9/lr_scale_acc.pdf), we performed hyperparameter tuning for deeper models and observed that, with appropriate learning rate adjustments, these models can successfully acquire analogical reasoning. This suggests that the lack of alignment in deeper models in our original setup is not a fundamental limitation of depth, but rather an artifact of suboptimal optimization.
>
> We will revise the wording to avoid the potentially misleading term *inverse scaling.* Importantly, our core mechanistic claim remains unchanged. We revise our interpretation to emphasize that compared to compositional reasoning, analogical reasoning is more sensitive to optimization and model scale, as it requires geometric alignment in the embedding space (Figure 7)
>
> ---
>
> **>W3**
> > A more direct test would be to intervene on the embedding geometry.
>
> To directly test whether the observed reduction in Dirichlet energy is causally linked to analogical reasoning, we conducted a layer-skipping intervention experiment in LLMs.
>
> Using 50 analogy examples from our toy task, we performed the following procedure:
>
> 1.Compute per-layer Dirichlet energy gradient ΔE(L) = E(L) − E(L−1) at functor-pair positions.
> 2. Identify the layer with the steepest energy drop (most negative ΔE).
> 3. Skip the layer (replace with identity mapping) and measure ΔP(target).
> 4. Compare against skipping randomly selected layers (averaged over 20 random seeds).
>
> |Top-1 ΔP | Random ΔP | Ratio |
> |----------|-----------|-------|
> |−0.055 ± 0.011 | −0.024 ± 0.007 | 2.3× |
>
> We find that skipping the layer most responsible for the energy reduction causes **2.3× greater degradation** in P(target) compared to skipping a random layer. This provides direct causal evidence that the reduction in Dirichlet energy plays a functional role in correct analogical prediction.
>
> Thank you for your helpful suggestion.
>
> ---
>
> **>W4**
> > It’s still unclear how well this synthetic setting applies to real-world analogy.
>
> To address this concern, we conducted additional experiments on a real-world natural language analogical reasoning dataset, **E-KAR** [1]. Details of this setup are described in our response to Reviewer **6fw4 (W1)**.
>
> The results (link: https://anonymous.4open.science/r/Emergent_Analogy_rebuttal-C3E9/dirichlet_eakr.pdf) show a similar trend to our toy setting.
>
> These results suggest that the mechanism identified in our toy setting generalizes to real-world analogical reasoning tasks.
>
> [1] Chen et al., *E-KAR: A Benchmark for Rationalizing Natural Language Analogical Reasoning,* Findings of ACL 2022.
>
> ---
>
> **>Q1**
> > Have the authors tried to run a variant of their experiment where this signal is absent or implicit?
>
> During the rebuttal period, we explored a variant of the task without an explicit functor token, following the suggestion by another reviewer. Please refer to the section  `[e1][e2][e3]->[e6][e7]` in our response to Reviewer **NfFL (W1)** for details.
>
> Our preliminary results suggest that analogical reasoning can still emerge even when the mapping signal is implicit, indicating that the model can infer latent correspondences from structural information.

---

> > ### Author Rebuttal · Reviewer_QSCk · 2026-04-02
> >
> > I thank the authors for their thorough rebuttal. The additional experiments directly address my concerns. This is a very strong paper and I am upgrading my score to Strong Accept.

---

> > > ### Author Response · Authors · 2026-04-04
> > >
> > > Thank you for your supportive feedback and the improved score.
> > > We are grateful for your insightful suggestions, which were instrumental in strengthening our work—particularly regarding the non-perfect isomorphism and causal interventions.
> > > We will ensure that these new results are fully integrated into the final manuscript.

---

### Official Review · Reviewer_6v9T · 2026-03-10

**Soundness:** 3
**Presentation:** 4
**Significance:** 2
**Originality:** 3
**Overall Recommendation:** 4
**Confidence:** 3

**Summary:**

The paper investigates the mechanisms by which Transformers acquire analogical reasoning, formalizing the problem as a category-theoretic functor mapping. Through controlled synthetic tasks, the authors demonstrate a three-stage training dynamic where analogical reasoning emerges last and is highly sensitive to dataset properties, optimization hyper-parameters, and model capacity. They decompose this capability into structural alignment within the embedding space and functor application via vector addition. The authors further use logit lens analysis to provide evidence that similar layer-wise structural alignment occurs in pretrained LLMs during in-context learning.

**Compliance With Llm Reviewing Policy:**

Affirmed.

**Final Justification:**

While the rebuttal provided helpful experiments, it revealed that the paper's core "inverse scaling" claim was merely an optimization artifact, compromising its original soundness. So I maintain my score.

**Key Questions For Authors:**

1. The synthetic task uses a single functor token $f$ for one mapping. Does the proposed mechanism generalize to multiple distinct functors concurrently, or does the model simply overfit to a single toy transformation?
2. Real-world analogy lacks explicit mapping cues. Can Transformers infer latent functors purely from structural isomorphism without injecting an explicit $f$ token during training?
3. Could the "inverse scaling" with depth be an artifact of using a constant learning rate ($1\times 10^{-4}$) across all sizes? Have you swept hyper-parameters for deeper models?
4. Given the extreme sensitivity to prompt formatting in Appendix H (e.g., Prompt 4 failing without <> markers), how can we confidently claim the LLM relies on abstract geometric alignment rather than shallow pattern matching?

**Limitations:**

yes

**Strengths And Weaknesses:**

Strengths:

- The formalization of analogy as a category-theoretic functor provides a clean, mathematically rigorous framework to isolate structural mapping from standard compositional reasoning.
- The mechanistic decomposition of reasoning into embedding structural alignment and attention-based retrieval is highly intuitive and backed by quantitative metrics like Dirichlet Energy and vector parallelism.

Weaknesses:

- The synthetic task relies on a single, explicit functor token $f$ and tests only one fixed cross-category mapping, severely restricting generalization and making the task too "toy" to reflect multi-relational, implicit analogies in the real world.
- The reported "inverse scaling" for depth and non-monotonic scaling for width are likely artifacts of sub-optimal, fixed hyper-parameter choices across varying model sizes, lacking sufficient ablation to claim fundamental architectural limitations.
- The LLM in-context learning evaluation is highly brittle and over-sensitive to surface-level formatting ( Appendix H), undermining the claim that deep structural alignment is driving the predictions.

---

> ### Author Rebuttal · Authors · 2026-03-31
>
> We thank the reviewer for the insightful comments, which helped improve the clarity and scope of our work.
>
> ---
>
> **>W1&Q1**
> > The synthetic task relies on a single, explicit functor token f and tests only one fixed cross-category mapping
>
> > Does the proposed mechanism generalize to multiple distinct functors concurrently, or does the model simply overfit to a single toy transformation?
>
> As noted in **Appendix L (Limitations)**, real-world settings involve multiple categories and multiple mappings (functors), whereas our main experiments focus on a simplified setting with a single functor. To address this concern, we conducted additional experiments with multiple functors.
>
> In these experiments (setup: https://anonymous.4open.science/r/Emergent_Analogy_rebuttal-C3E9/multi_f.pdf), we extend the setup to include multiple categories and corresponding functor tokens. For example, a functor $f_1$ maps between Category 0 and Category 1, while $f_2$ maps between Category 0 and Category 2, and so on. We fix the total number of entities to 100 and vary the number of categories: 2 categories (50:50, as in the main text), 4 categories (25×4), and 5 categories (20×5).
>
> The results (link: https://anonymous.4open.science/r/Emergent_Analogy_rebuttal-C3E9/multi_f_acc.pdf) show that analogical reasoning consistently emerges across all settings. The emergence is fastest in the case with 5 categories, which we attribute to the smaller number of entities per category (consistent with the trends observed in Figure 2).
>
> These results suggest that our findings are not limited to a single toy transformation: even when multiple functors are introduced, the same qualitative behavior is preserved.
>
> We will include these additional experiments in the appendix. Thank you again for the helpful suggestion.
>
> ---
>
> **>W3&Q4**
> > The LLM in-context learning evaluation is highly brittle and over-sensitive to surface-level formatting ( Appendix H)
>
> > how can we confidently claim the LLM relies on abstract geometric alignment rather than shallow pattern matching?
>
> To address the brittleness observed in Appendix H, we conducted additional experiments on a real-world natural language analogical reasoning dataset, **E-KAR** [1]. Details of this setup are described in our response to Reviewer 6fw4 (W1).
>
> The results (link: https://anonymous.4open.science/r/Emergent_Analogy_rebuttal-C3E9/dirichlet_eakr.pdf) show a similar trend to our toy setting: the Dirichlet energy corresponding to geometric alignment decreases across layers, and this decrease is followed by an increase in the logit-lens probability of the correct answer in later layers.
>
> These findings suggest that the observed mechanism is not solely driven by surface-level formatting or shallow pattern matching, but also appears in more naturalistic analogical reasoning tasks.
>
> We will include these additional experiments in the appendix.
>
> [1] Chen et al., *E-KAR: A Benchmark for Rationalizing Natural Language Analogical Reasoning,* Findings of ACL 2022.
>
> ---
>
> **>Q2**
> >  Real-world analogy lacks explicit mapping cues. Can Transformers infer latent functors purely from structural isomorphism without injecting an explicit f token during training?
>
> As discussed in **Section 6**, using an explicit functor token is a simplification of our setting.
>
> During the rebuttal period, we explored the setting proposed by Reviewer NfFL, where no explicit functor token is used. Please refer to the section  `[e1][e2][e3]->[e6][e7]` in our response to Reviewer **NfFL (W1)** for details.
>
> Our preliminary results suggest that analogical reasoning can emerge even without an explicit $f$ token, indicating that Transformers may infer latent mappings from structural information alone.
>
> ---
>
> **>W2&Q3**
> > The reported "inverse scaling"
>
> > Have you swept hyper-parameters for deeper models?
>
> We agree that the observed “inverse scaling” may be influenced by suboptimal hyperparameter choices. In our initial experiments, we used a fixed learning rate (1×10⁻⁴) across all model sizes, which is likely not optimal for larger or deeper models.
>
> In additional experiments (https://anonymous.4open.science/r/Emergent_Analogy_rebuttal-C3E9/lr_scale_acc.pdf), we performed learning rate sweeps for deeper models (e.g., $d_{\mathrm{model}}=512$ and $n_{\mathrm{layer}}=4$) and found that, with appropriate tuning, these models can achieve strong analogical reasoning performance (up to ~90%). This suggests that the observed “inverse scaling” is at least partly an optimization artifact rather than a fundamental architectural limitation.
>
> We will revise the wording to avoid the potentially misleading term *inverse scaling.* That said, our overall claim remains unchanged. We revise our interpretation to emphasize that analogical reasoning is more sensitive to optimization and model scaling (Figure 3,4) because it requires geometric alignment in the embedding space, as illustrated in Figure 7.
>
> We thank the reviewer for this helpful suggestion.

---

> > ### Author Rebuttal · Reviewer_6v9T · 2026-04-04
> >
> > I appreciate the authors' extensive efforts during the rebuttal, particularly the multi-functor experiments and learning rate sweeps. However, my core concerns remain. The rebuttal reveals that the "inverse scaling" phenomenon—a important claim of the original manuscript—was largely an optimization artifact rather than a fundamental architectural limitation. Reversing such a critical finding requires a substantial rewrite of Section 3.3 and fundamentally alters the paper's core narrative.

---

> > > ### Author Response · Authors · 2026-04-05
> > >
> > > Thank you for your rigorous follow-up. We are pleased that our rebuttal and additional experiments have addressed many of your initial concerns. Regarding the scaling of deeper models, we appreciate your insight, which helped us clarify that those findings were an optimization artifact rather than a fundamental architectural limit.
> > >
> > > However, we would like to emphasize that "inverse scaling" was never our core narrative. As you highlighted in the "Strengths," the primary contributions of this work remain (1) the formalization via category theory, (2) the analysis of training dynamics, and (3) the mechanistic discovery of analogical reasoning.
> > > In our revision of Section 3.3, we will replace the potentially misleading term "inverse scaling" and refine our interpretation to emphasize that analogical reasoning is uniquely sensitive to optimization because it requires precise geometric alignment in the embedding space. Since the depth issue was just one empirical observation, we believe that a minor refinement to the tone of our discussion will sufficiently address this concern.
> > >
> > > We are grateful for your constructive feedback, which has ultimately helped us present a more accurate and robust account of our findings.

---

### Official Review · Reviewer_NfFL · 2026-03-13

**Soundness:** 3
**Presentation:** 4
**Significance:** 2
**Originality:** 3
**Overall Recommendation:** 5
**Confidence:** 4

**Summary:**

This paper studies whether and how analogical reasoning emerges in Transformers. Analogical reasoning is defined as the ability to find a cross-domain function f that maps entities e_1, ... e_N to another domain, such that f(e_1), ..., f(e_N) share the same relations as e_1, ... e_N.

The authors find that unlike compositional reasoning (which can be learned consistently), the emergence of analogical reasoning is sensitive to training conditions. The paper also provides a mechanistic explanation for how transformers perform analogical reasoning: 1) the embedding of different domains become similarity structured through training. 2) the transformer applies vector addition to map from e_s to e_t. They further show that pre-trained LLMs exhibit similar behavior along the layer axis.

**Compliance With Llm Reviewing Policy:**

Affirmed.

**Final Justification:**

The authors' rebuttal have addressed all of my concerns.

**Key Questions For Authors:**

- Figure 2 shows that different OOD ratio leads to similar training behaviors for memorization and compositional reasoning. Is the OOD ratio applied only to the training data, while all test data are OOD?
- Figure 6 (a): Dirichlet energy does not decrease monotonically with increasing accuracy. Could the authors provide intuition for why this might occur?
- Figure 9. has a sudden increase in the prediction accuracy of e_t. p(e_t) is determined using logit lens. Although widely used, it may not accurately reflect intermediate predictions. Earlier layers may encode information differently than the final linear readout layer can decode. It may be more reliable to train probes for internal states of "what the output entity should be".
- Does Figure 15 in Appendix G show the PCA result for successful analogical reasoning? If so, this may further suggest that vector addition alone cannot fully explain the embedding structure.

Minor issues: The font size in Figure 5 is too small.

**Limitations:**

Yes. Section 6 (Sample Efficiency and Creativity) and Appendix L.

**Strengths And Weaknesses:**

# Strengths
- Anological reasoning is an important aspect of symbolic reasoning that enables OOD generalization. For solving a math problem, compositional reasoning allows the model to follow a know reasoning template, while analogical reasoning finds the right reasoning template to use from existing samples, which potentially improves sample-efficiency. Understanding analogical reasoning may guide future models with better problem-solving skills and creativity.
- The paper is very well-structured and clearly written. In particular, Figure 1 provides a clear and intuitive overview of the main idea of the paper.
- I appreciate how the OOD ratio is controlled for both compositional reasoning and analogical reasoning, which allows for a direct comparison between the two during training.


# Weaknesses
- The proposed anological reasoning task may not truly require analogical reasoning:
    + Here, I want to argue that if the model has perfect compositional reasoning and inverse-relation reasoning, i.e. it knows [r53][r35]=identity, then it can solve the proposed analogical reasoning task.
    + Consider the exact graph as in Fig.1(A), the query [e5][f] can be solved by inserting an identity relationsihp: =[e5][r53][f][r35]. Given that [r8,10]=[r35](this is explicitly given, the same relation token is used), [e5][f]=[e3][r53][f][r8,10]=[e3][f][r8,10]=[e8][r8,10]=[e10]
    + Under the authors' setup, the functor f could also be interpreted as a relationship, just as r_ij. If this interpretation holds, the proposed analogical reasoning task effectively reduces to multi-hop compositional reasoning.
    + One alternative that may better reflect analogical reasoning is [e1][e2][e3]->[e6][e7]?, where the model needs to infer <e8>. This framing better aligns with how analogies are used in the real-world: it's unlikely a complete set of entities in a domain matches the same relationships as another. The mapping is often partial.

- The mechanistic explanation (Section 4) is a bit disconnected from the training dynamics of analogical reasoning discussed in Secion 3. A mechanistic explanation should ideally help explain why analogical reasoning is hard: One hypothesis is that analogical reasoning requires embeddings from different domains to be placed in similarly structured regions of the embedding space, which may be more brittle to changes.
- The mechanistic explanation is not fully convincing. The evidence appears to rely on correlation instead of causation, and the conclusions are based on limited evidence.
    + The main evidence for vector addition is the observed correlation between analogical reasoning performance and attention scores (e_s->f), as well as sim(e_t-e_s,f). Correlation alone does not support causation. Causal mediation used as in [1][2] could provide stronger evidence, for example by patching the activations of a different e_s/ f.
    + Parallelism (i.e., cosine similariy between (e_t-e_s, f)) is low in value (~0.15)
- More generally, the family of functions that relate e_s and e_t may not necessarily be additive. Recent papers [3][4] suggest that a robust relational representation may be bilinear, of the form: IsRelationship(e_s,e_t) = e_x^TWe_t (which resembles the qudratic interactions used in self-attention in transformers). Under this asssumption, analogical consistency would require e_s^TWe_t = f(e_s)^T⋅W⋅f(e_t). If f(x) is linear, i.e., f(x)=Ax+b, Wb=0 and A^TWA=W. Vector addition (e_t=e_s+f) then becomes a special case that satisfies these constraints, where f should be orthogonal to W. This hypothesis can be tested through training bilinear probes [4] to model relationships between entities and then checking if Wf=0 for each relationship. This could provide a better geometric understanding of the embedding structure that enables analogical reasoning.


References:

[1] Locating and Editing Factual Associations in GPT: https://arxiv.org/pdf/2202.05262;

[2] How do Language Models Bind Entities in Context?: https://arxiv.org/abs/2310.17191;

[3] Bilinear representation mitigates reversal curse and enables consistent model editing: https://arxiv.org/abs/2509.21993;

[4] Does Object Binding Naturally Emerge in Large Pretrained Vision Transformers?: https://arxiv.org/abs/2510.24709.

---

> ### Author Rebuttal · Authors · 2026-03-31
>
> We thank the reviewer for the insightful comments and helpful suggestions.
>
> ---
>
> **>W1**
> > anological reasoning task may not truly require analogical reasoning:
>
> We understand your concern as questioning whether our task reduces to compositional reasoning.
>
> We would like to respond from several perspectives:
>
> (1) Simplified illustrative case.
>
> While your example is intuitive, our setting is more complex. In harder regimes (e.g., analogical ratio OOD in Figure 2 or missing relations in Appendix C), such decomposition becomes less straightforward, and the model relies more on global structural reasoning.
>
> (2) Definition of compositional reasoning.
>
> Compositional reasoning combines observed atomic facts e.g., e1,r12,e2 and e2,r23,e3, to infer e1,r12,r23,e3. In contrast, analogical reasoning only observes e,f, without intermediate chains, making the reaosning fundamentally different.
>
> (3) Distinct dynamics and mechanisms.
>
> We observe different training dynamics, suggesting analogy is not merely compositional reuse.
>
> (4) Consistency with real-world data.
>
> Most importantly, similar patterns appear in natural language analogy benchmarks (Reviewer **6fw4, W1**), supporting that our task captures a meaningful abstraction of analogy.
> > [e1][e2][e3]->[e6][e7]
>
> We find your suggestion, removing the explicit functor, interesting and well-motivated.
>
> To explore this, we conducted additional experiments comparing our original setting with your proposed formulation (e1,e2,e3,→e6,e7,e8). For clarity, we use → notation here, but no explicit token is introduced.
>
> The result (link: https://anonymous.4open.science/r/Emergent_Analogy_rebuttal-C3E9/no_functor_acc.pdf) shows that both settings exhibit similar learning dynamics, although the proposed setting is more challenging, with performance saturating around 90%.
>
> These results suggest that analogical reasoning can emerge even without an explicit functor token, mitigating the concern that the task can be reduced to compositional reasoning.
>
> ---
> **>W2**
> > A mechanistic explanation
>
> As you pointed out, and as shown in Figure 7, the difficulty of analogical reasoning arises from the need for geometric alignment.
>
> We will clarify this explanation.
>
> ---
> **>W3&W4**
> > rely on correlation instead of causation
>
> >  checking if Wf=0.
>
> To address causation, we conduct additional causal intervention experiments (see **QSCk W3**).
>
> We also explored a bilinear probe[1], which probes whether a pair of entities ($e_s, e_t$) is in a relation by modeling it as $e_s^T W e_t$, achieving 81.1% accuracy.
>
> We then analyze the interaction between $W$ and the functor direction $f$ by tracking $\frac{\|Wf\|}{1/N\sum_i^N \|We_i\|}$. As shown in figure (https://anonymous.4open.science/r/Emergent_Analogy_rebuttal-C3E9/Wf.pdf), this value decreases during training, from 0.624 to 0.108, indicating that $f$ progressively aligns with the null space of $W$.
>
> We further perform a permutation test: we sample 1,000 random directions $v$ and compare $\|Wv\|$ to $\|Wf\|$. We find that 98.2% of random directions have larger $\|Wv\|$ than $|Wf\|$ ($p = 0.018$), confirming that $f$ is a statistically special direction with minimal relational interaction.
>
> These results suggest that our hypothesis can also be validated under the bilinear formulation, by checking whether $Wf \approx 0$.
>
> [1] *Bilinear representation mitigates reversal curse and enables consistent model editing.*
>
> ---
> **>Q1**
> > Is the OOD ratio applied only to the training data, while all test data are OOD?
>
> As described in Section 2, the OOD ratio specifies the fraction of samples held out from training; all evaluation samples are OOD by construction.
>
> ---
> **>Q2**
> > Dirichlet energy does not decrease monotonically
>
> With cross-entropy training, embedding norms can keep increasing after loss convergence. Since Dirichlet energy depends on distances, this can slightly increase energy even after accuracy saturates.
>
> See (link: https://anonymous.4open.science/r/Emergent_Analogy_rebuttal-C3E9/energy_norm.pdf).
>
> ---
> **>Q3**
> > It may be more reliable to train probes
>
> Layer-wise logit lens analysis is widely used [3,4], though it may not accurately decode early-layer representations.
>
> We conducted additional experiments using linear probing on Gemma2-9B. The result (link: https://anonymous.4open.science/r/Emergent_Analogy_rebuttal-C3E9/linearprobe.pdf) shows a similar trend: performance improves as Dirichlet energy decreases, with linear probes rising earlier than logit lens.
>
> [2] Lad et al., *The Remarkable Robustness of LLMs: Stages of Inference?*, NeurIPS 2025
>
> [3] Shao et al., *Benford’s Curse: Tracing Digit Bias to Numerical Hallucination in LLMs*, NeurIPS 2025
>
> ---
> **>Q4**
> > Figure 15
>
> Figure 15(b), which corresponds to the 4-layer Transformer, shows the case where analogical reasoning fails.
>
> ---
> **>Q5**
> > font size in Figure 5 is too small.
>
> We updated Figure 5: https://anonymous.4open.science/r/Emergent_Analogy_rebuttal-C3E9/rebuttal_fig5.pdf.

---

> > ### Author Rebuttal · Reviewer_NfFL · 2026-04-04
> >
> > I appreciate the authors’ detailed rebuttal and additional experiments. They have addressed all of my concerns, and I am raising my score. I look forward to the final manuscript.

---

> > > ### Author Response · Authors · 2026-04-04
> > >
> > > Thank you for your time and for the positive update to your evaluation.
> > > We are especially grateful for your insightful suggestions regarding the bilinear formulation and the implicit functor task.
> > > These additional analyses have significantly strengthened the theoretical and empirical grounding of our work, and we will ensure they are fully reflected in the final manuscript.

---

### Official Review · Reviewer_6fw4 · 2026-03-13

**Soundness:** 3
**Presentation:** 4
**Significance:** 3
**Originality:** 4
**Overall Recommendation:** 5
**Confidence:** 4

**Summary:**

In this paper, the authors explore the presence of analogical reasoning in Transformer models. They formalize the concept of analogies as "functors" (isomorphic mappings between relational structures). They investigate whether analogical reasoning—i.e., the correct application of functors to out-of-distribution (OOD) examples—can emerge in a synthetic-data task, beyond memorization and compositional reasoning.

They find that the training dynamics follow a three-stage progression in which different forms of reasoning (memorization, compositional reasoning, and analogical reasoning) emerge gradually. The authors also reveal how parameters of the synthetic data influence these training dynamics. In addition, they analyze the effects of optimization settings such as weight decay, batch size, model width, and number of layers.

The paper further examines the mechanism underlying analogy. The authors show experimentally that the subjects of analogies are embedded in a structurally aligned manner, and that functor application is implemented as vector addition within Transformer layers. Finally, they conduct experiments with a real large language model in an in-context learning setting.

**Compliance With Llm Reviewing Policy:**

Affirmed.

**Final Justification:**

The authors have addressed my questions satisfactorily and strengthened the paper by connecting their findings to real-world examples through additional experiments, which I consider an important improvement. Overall, the paper provides important insights into model behavior and offers significant value to the research community. As I believe this remains the appropriate assessment, I keep my overall rating at Accept.

**Key Questions For Authors:**

The authors state: “As shown in Figure 6(b), this attention score (purple) increases at the same time analogical reasoning performance improves” (p. 6, col. 2, line 322). However, the plot appears to show a slightly different pattern: the attention score begins to increase after the rise in analogical reasoning performance. What could explain this discrepancy?

What causes wider models ($d_{model} = 512$) to fail to learn analogical reasoning? (p. 5, col. 1, line 262)

In the section on analogies in LLMs, the authors find that the phenomenon manifests along the depth (layer) axis (p. 8, col. 1, line 395). Does the same pattern also appear in the toy model?

**Limitations:**

Yes.

**Strengths And Weaknesses:**

Strengths

- The paper addresses an interesting and underexplored problem, and introduces a clean synthetic framework that separates memorization, compositional reasoning, and analogical reasoning.
- The paper is well written and enjoyable to read.
- The exploration of data settings and optimization parameters is comprehensive, and the paper makes a good-faith effort to connect toy-model findings to pretrained LLMs, including prompt designs intended to reduce trivial lexical or numerical shortcuts.
- The analysis of the underlying mechanism uses an insightful concept (Dirichlet energy) that provides a clear description of the latent structure of the representations.

Overall, the paper presents important empirical findings and offers a formal framework for investigating analogical reasoning in Transformers.

Weaknesses

- The paper does not clearly establish the connection between real-world analogical reasoning and the formalized algebraic framework used in the study. Additionally, the LLM section relies only on synthetic examples similar to those used in the toy model.

---

> ### Author Rebuttal · Authors · 2026-03-31
>
> We thank the reviewer for the thoughtful comments and valuable suggestions.
>
> ---
>
> **>W1**
> > The paper does not clearly establish the connection between real-world analogical reasoning
>
> As noted in **Appendix L (Limitations)**, connecting our findings to real-world analogical reasoning remains an important challenge.
> To address this concern, we conducted additional experiments on a real-world analogical reasoning benchmark in natural language processing, **E-KAR** [1], in order to examine whether our key findings, the decrease in Dirichlet energy and the improvement in analogical reasoning performance, also hold beyond the synthetic setting.
> E-KAR is a multiple-choice analogy dataset. An example is shown below:
> ```
> engine:car :: A) ink:brush B) moonlight:sky C) sailboat:sea D) wings:butterfly
> Answer:
> ```
> This task requires identifying a pair that shares the same relation as engine:car. Since engine *is part of* car and wings *are part of* butterfly, the correct answer is *D*. The dataset contains 868 such examples.
>
> Using Gemma2-9B, we performed an analysis analogous to our main experiments. Specifically, we measured the Dirichlet energy between entity pairs (e.g., engine:car and wings:butterfly) across layers, and also tracked the logit-lens probability of the correct answer (e.g., D). We observe that, as depth increases, the Dirichlet energy consistently decreases, and this decrease coincides with an increase in the probability assigned to the correct answer (see link: https://anonymous.4open.science/r/Emergent_Analogy_rebuttal-C3E9/dirichlet_eakr.pdf). These results suggest that the mechanism identified in our toy setting generalizes to real-world analogical reasoning tasks.
>
> We appreciate the reviewer’s valuable suggestion, which helped strengthen our empirical validation.
>
> [1] Chen et al., *E-KAR: A Benchmark for Rationalizing Natural Language Analogical Reasoning,* Findings of ACL 2022.
>
> ---
>
> **>Q1**
> > the attention score begins to increase after the rise in analogical reasoning performance. What could explain this discrepancy?
>
> We agree that the phrasing “increases at the same time” is somewhat misleading.
> As shown in Figure 6(b), the attention score actually begins to increase before the rise in analogical reasoning probability. This pattern is consistent with the intuition that internal circuit formation or mechanistic signals can precede observable changes in model outputs. Such behavior is commonly reported in interpretability research. For example, prior work on grokking has shown that internal progress measures change before improvements in generalization performance become visible [2]. Our observations are aligned with this phenomenon.
>
> We will revise the statement from
> “attention score (purple) increases at the same time analogical reasoning performance improves” to “the attention score increases prior to improvements in analogical reasoning performance,” and include the above discussion for clarification.
>
> [2] Nanda et al., *Progress measures for grokking via mechanistic interpretability,* ICLR 2023.
>
> ---
>
> **>Q2**
> > What causes wider models (d_model=512) to fail to learn analogical reasoning?
>
> As also noted by Reviewer 6v9T, during the rebuttal period we found that larger models in our toy setting require careful tuning of the learning rate.
>
> In particular, as shown in (https://anonymous.4open.science/r/Emergent_Analogy_rebuttal-C3E9/lr_scale_acc.pdf), when using $d_{\mathrm{model}}=512$ and $n_{\mathrm{layer}}=4$, adjusting the learning rate enables the model to achieve nearly 90% accuracy on analogical reasoning.
>
> Therefore, the statement that larger models **fail** to learn analogical reasoning is somewhat misleading. Our main claim remains unchanged; however, we will revise the wording to reflect that analogical reasoning is more **sensitive** to optimization and model size than compositional reasoning, as it requires geometric alignment in the embedding space (Figure 7).
>
> ---
> **>Q3**
> > Does the same pattern also appear in the toy model?
>
> As noted in the paper (p. 8, line 396), the layer-wise decrease in Dirichlet energy observed in LLMs can be attributed to the fact that LLMs perform in-context learning, where representations are progressively refined across layers. Prior work has shown that this process can be interpreted as a form of optimization unfolding along the depth of the network [3].
> In contrast, the toy model in our study is *not* designed to evaluate in-context learning. Instead, representation learning occurs through weight updates during training. Therefore, the key phenomenon in the toy model appears along the training-step axis rather than the layer axis, and layer-wise variation is not essential to the mechanism we analyze.
>
> [3] Von Oswald et al., *Transformers Learn In-Context by Gradient Descent.*, ICML2023

---

> > ### Author Rebuttal · Reviewer_6fw4 · 2026-04-04
> >
> > I thank the authors for their responses. I particularly appreciate the additional result on real-world analogical reasoning, which I found very valuable. All of my questions have been fully addressed. I look forward to the final version of the manuscript.

---

> > > ### Author Response · Authors · 2026-04-04
> > >
> > > Thank you for your thoughtful feedback and for acknowledging our efforts. We are particularly glad that the additional results on real-world analogical reasoning (E-KAR) proved valuable to you. Your suggestion was instrumental in demonstrating the generality of our proposed mechanism.

---

### Decision · Program_Chairs · 2026-04-30

**Decision:**

Accept (spotlight)

**Comment:**

This paper received strong support from reviewers on an interesting topic: relating the dynamics of learning to make analogies in transformer models to a formalization of analogy via category theory. The author response improved the breadth of the empirical results by including a test of the explanation on an open-weights large language model. Though this work does not provide prescise theory on the mechanism of the emergence as it is an empirical paper with some descriptive analyses, the setting appears ripe for further investigation.

*Minor*: The authors should attribute Chan et al. (2022) as an example of tying emergent complex behavior to data distributional properties. Presently, they are only attributed as using synthetic tasks to probe models.